# *GOAt*: Explaining Graph Neural Networks via Graph Output Attribution

**Shengyao Lu[1], Keith G. Mills[1], Jiao He[2], Bang Liu[3], Di Niu[1]**
[1]Department of Electrical and Computer Engineering, University of Alberta
[2]Kirin AI Algorithm & Solution, Huawei  [3]DIRO, Université de Montréal & Mila
{shengyao,kgmills,dniu}@ualberta.ca
hejiao4@huawei.com,bang.liu@umontreal.ca

## Abstract

Understanding the decision-making process of Graph Neural Networks (GNNs) is crucial to their interpretability. Most existing methods for explaining GNNs typically rely on training auxiliary models, resulting in the explanations remain black-boxed. This paper introduces Graph Output Attribution (GOAt), a novel method to attribute graph outputs to input graph features, creating GNN explanations that are faithful, discriminative, as well as stable across similar samples. By expanding the GNN as a sum of scalar products involving node features, edge features and activation patterns, we propose an efficient analytical method to compute contribution of each node or edge feature to each scalar product and aggregate the contributions from all scalar products in the expansion form to derive the importance of each node and edge. Through extensive experiments on synthetic and real-world data, we show that our method not only outperforms various state-of-the-art GNN explainers in terms of the commonly used fidelity metric, but also exhibits stronger discriminability, and stability by a remarkable margin. Code can be found at: https://github.com/sluxsr/GOAt.

## 1 Introduction

Graph Neural Networks (GNNs) have demonstrated notable success in learning representations from graph-structured data in various fields (Kipf & Welling, 2017; Hamilton et al., 2017; Xu et al., 2019). However, their black-box nature has driven the need for explainability, especially in sectors where transparency and accountability are essential, such as finance (Wang et al., 2021), healthcare (Amann et al., 2020), and security (Pei et al., 2020). The ability to interpret GNNs can provide insights into the mechanisms underlying deep models and help establish trustworthy predictions.

Existing attempts to explain GNNs usually focus on local-level or global-level explainability. Local-level explainers (Ying et al., 2019; Luo et al., 2020; Schlichtkrull et al., 2021; Vu & Thai, 2020; Huang et al., 2022; Lin et al., 2021; Shan et al., 2021; Bajaj et al., 2021) typically train a secondary model to identify the critical graph structures that best explain the behavior of a pretrained GNN for specific input instances. Global-level explainers (Azzolin et al., 2023; Huang et al., 2023) perform prototype learning or random walk on the explanation instances to extract the global concepts or counterfactual rules over a multitude of graph samples. While global explanations can be robust, local explanations are more intuitive and fine-grained.

In this paper, we introduce a computationally efficient local-level GNN explanation technique called **G**raph **O**utput **At**tribution (*GOAt*) to overcome the limitations of existing methods. Unlike methods that rely on back-propagation with gradients (Pope et al., 2019; Baldassarre & Azizpour, 2019; Schnake et al., 2021; Feng et al., 2022) and those relyinig on hyper-parameters or training complex black-box models (Luo et al., 2020; Lin et al., 2021; Shan et al., 2021; Bajaj et al., 2021), our approach enables attribution of GNN output to input features, leveraging the repetitive sum-product structure in the forward pass of a GNN.

Given that the matrix multiplication in each GNN layer adheres to linearity properties and the activation functions operate element-wise, a GNN can be represented in an expansion form as a sum of

scalar product terms, involving input graph features, model parameters, as well as *activation patterns* that indicate the activation levels of the scalar products. Based on the notion that all scalar variables $X_i$ in a scalar product term $g = cX_1X_2 \ldots X_N$ contribute equally to $g$, where $c$ is a constant, we can attribute each product term to its corresponding factors and thus to input features, obtaining the importance of each node or edge feature in the input graph to GNN outputs. We present case studies that demonstrate the effectiveness of our analytical explanation method *GOAt* on typical GNN variants, including GCN, GraphSAGE, and GIN.

Besides the fidelity metric, which is commonly used to assess the faithfulness of GNN explanations, we introduce two new metrics to evaluate the *discriminability* and *stability* of the explanation, which are under-investigated by prior literature. Discriminability refers to the ability of explanations to distinguish between classes, which is assessed by the difference between the mean explanation embeddings of different classes, while stability refers to the ability to generate consistent explanations across similar data instances, which is measured by the percentage of data samples covered by top-$k$ explanations. Through comprehensive experiments based on on both synthetic and real-world datasets along with qualitative analysis, we show the outstanding performance of our proposed method, *GOAt*, in providing highly faithful, discriminative, and stable explanations for GNNs, as compared to a range of state-of-the-art methods.

## 2 Problem Formulation

**Graph Neural Networks.** Let $G = (\mathcal{V}, \mathcal{E})$ be a graph, where $\mathcal{V} = \{v_1, v_2, \ldots, v_N\}$ denotes the set of nodes and $\mathcal{E} \subseteq \mathcal{V} \times \mathcal{V}$ denotes the set of edges. The node feature matrix of the graph is represented by $X \in \mathbb{R}^{N \times d}$, and the adjacency matrix is represented by $A \in \{0,1\}^{N \times N}$ such that $A_{ij} = 1$ if there exists an edge between nodes $v_i$ and $v_j$. The task of a GNN is to learn a function $f(G)$, which maps the input graph $G$ to a target output, such as node labels, graph labels, or edge labels. Formally speaking, for a given GNN, the hidden state $h_i^{(l)}$ of node $v_i$ at its layer $l$ can be represented as:

$$h_i^{(l)} = \text{COMBINE}^{(l)} \left\{ h_i^{(l-1)}, \text{AGGREGATE}^{(l)} \left( \left\{ h_j^{(l-1)}, \forall v_j \in \mathcal{N}_i \right\} \right) \right\}, \tag{1}$$

where $\mathcal{N}_i$ represents the set of neighbors of node $v_i$ in the graph. $\text{COMEBINE}^{(l)}(\cdot)$ is a COMBINE function such as concatenation (Hamilton et al., 2017), while $\text{AGGREGATE}^{(l)}(\cdot)$ are AGGREGATE functions with aggregators such as ADD. We focus on GNNs that adopt the non-linear activation function ReLU in COMBINE or AGGREGATE functions.

**Local-level GNN Explainability.** Our goal is to generate a faithful explanation for each graph instance $G = (\mathcal{V}, \mathcal{E})$ by identifying a subset of edges $S \subseteq \mathcal{E}$, which are important to the predictions, given a GNN $f(\cdot)$ pretrained on a set of graphs $\mathcal{G}$. We highlight edges instead of nodes as suggested by (Faber et al., 2021) that edges have more fine-grained information than nodes while giving human-understandable explanations like subgraphs.

## 3 Method

The fundamental idea of our approach can be summarized as two key points: i) Equal Contribution of product terms; ii) Expanding the output representation to product terms. Before diving into the mathematical definitions, we briefly describe them by toy examples as following.

**Equal Contribution.** Given a product term $z = 10A_{11}A_{12}A_{23}$, where the variables $A_{11} = A_{12} = A_{23} = 1$ indicate that all three edges exist, resulting in $z = 10$. If any of the edges is missing, i.e., at least one of $A_{11}, A_{12}, A_{23}$ is 0, then $z = 0$. This fact implies that the presence of all edges is equally essential for the resulting value of $z = 0$. Therefore, each of the three edges contributes $1/3$ to the output, resulting in an attribution of $10/3$.

**Expand the output representation.** The output matrix of a GNN can be described as the outcome of a linear transformation involving the input matrices and the GNN parameters. Since the GNNs are pretrained, the parameters $W, B$ are fixed, hence can be treated as constants. As a result, each element within the output matrix can be represented as the sum of scalar products that involve entries from the inputs. Consequently, we can determine the attribution of an edge, by summing its contribution across all the scalar products in which it participates. For example, if the expansion

form of an element in the output matrix is $y_{13} = 10A_{11}A_{12}A_{23} + 8A_{11}A_{13}A_{33} + 11A_{12}A_{21}A_{13}$, then the attribution of $A_{11}$ to $y_{13}$ is $10/3 + 8/3 = 6$ since $A_{11}$ does not participate in the term $11A_{12}A_{21}A_{13}$.

The rest of this section begins by presenting our fundamental definition of equal contribution in a product term. Then, we mathematically present *GOAt* method for explaining typical GNNs, followed by a case study on GCN (Kipf & Welling, 2017). Additional case studies of applying *GOAt* to GraphSAGE (Hamilton et al., 2017) and GIN (Xu et al., 2019) are included in Appendix E and D. The code implementing our algorithm is included in the Supplementary Material of our submission.

## 3.1 EQUAL CONTRIBUTION

**Definition 1 (Equal Contribution).** *Given a function $g(\mathbf{X})$ where $\mathbf{X} = \{X_1, \ldots, X_M\}$ represents $M$ variables, we say that variables $X_i$ and $X_j$ have equal contribution to function $g$ at $(x_i, x_j)$ with respect to the base manifold at $(x'_i, x'_j)$ if and only if setting $X_i = x_i, X_j = x'_j$ and setting $X_i = x'_i, X_j = x_j$ yield the same manifold, i.e.,*

$$g_{X_i=x_i, X_j=x'_j}(\mathbf{X}\backslash\{X_i, X_j\}) = g_{X_i=x'_i, X_j=x_j}(\mathbf{X}\backslash\{X_i, X_j\})$$

*for any values of $\mathbf{X}$ excluding $X_i$ and $X_j$.*

**Lemma 2 (Equal Contribution in a product).** *Given a function $g(\mathbf{X})$ defined as $g(\mathbf{X}) = b\prod_{k=1}^{M} X_k$, where $b$ is a constant, and $\mathbf{X} = \{X_1, \ldots, X_M\}$ represents $M$ uncorrelated variables. Each variable $X_k$ is either $0$ or $x_k$, depending on the absence or presence of a certain feature. Then, all the variables in $\mathbf{X}$ contribute equally to $g(\mathbf{X})$ at $[x_1, \ldots, x_M]$ with respect to $[0, \ldots, 0]$.*

Proofs of all Lemmas and Theorems can be found in the Appendix. Since all the binary variables have equal contribution, we define the contribution of each variable $X_k$ to $g(\mathbf{X}) = b\prod_{k=1}^{M} X_k$ for all $k = 1, \ldots, M$, as

$$I_{X_k} = \frac{b\prod_{i=1}^{M} x_i}{M}. \tag{2}$$

## 3.2 EXPLAINING GRAPH NEURAL NETWORKS VIA ATTRIBUTION

A typical GNN (Kipf & Welling, 2017; Hamilton et al., 2017; Xu et al., 2019) for node or graph classification tasks usually comprises 2-6 message-passing layers for learning node or graph representations, followed by several fully connected layers that serve as the classifier. With the hidden state $h_i^{(l)}$ of node $v_i$ at the $l$-th message-passing layer defined as Equation (1), we generally have the hidden state $H^{(l)}$ of a data sample as:

$$H^{(l)} = \sigma\left(\Phi^{(l)}\left(\left(A + \epsilon^{(l)}I\right)H^{(l-1)}\right) + \lambda\Psi^{(l)}\left(H^{(l-1)}\right)\right), \tag{3}$$

where $A$ is the adjacency matrix, $\epsilon^{(l)}$ refers to the self-loop added to the graph if fixed to 1, otherwise it is a learnable parameter, $\sigma(\cdot)$ is the element wise activation function, $\Phi^{(l)}$ and $\Psi^{(l)}$ can be Multilayer Perceptrons (MLP) or linear mappings, $\lambda \in \{0, 1\}$ determines whether a concatenation is required. If the COMBINE step of a GNN requires a concatenation, we have $\lambda = 1$ and $\epsilon^{(l)} = 1$; if the COMBINE step requires a weighted sum, we have $\epsilon^{(l)}$ set trainable and $\lambda = 0$. Alternatively, Equation (3) can be expanded to:

$$H^{(l)} = \sigma\left(AH^{(l-1)}\prod_{k=1}^{K} W^{\Phi_k^{(l)}} + \epsilon^{(l)}H^{(l-1)}\prod_{k=1}^{K} W^{\Phi_k^{(l)}} + \lambda H^{(l-1)}\prod_{q=1}^{Q} W^{\Psi_q^{(l)}}\right), \tag{4}$$

where $K, Q$ refer to the number of MLP layers in $\Phi^{(l)}(\cdot)$ and $\Psi^{(l)}(\cdot)$, and $W^{\Phi_k^{(l)}}$ and $W^{\Psi_q^{(l)}}$ are the trainable parameters in $\Phi_k^{(l)}$ and $\Psi_q^{(l)}$.

Given a certain data sample and a pretrained GNN, for an element-wise activation function we can define the activation pattern as the ratio between the output and input of the activation function:

**Definition 3 (Activation Pattern).** *Denote $H^{(l)\prime}$ and $H^{(l)}$ as the hidden representations before and after passing through the element-wise activation function at the $l$-th layer, we define activation*

pattern $P^{(l)}$ for a given data sample as

$$P_{i,j}^{(l)} = \begin{cases} \frac{H_{i,j}^{(l)}}{H_{i,j}^{(l)'}}, & \text{if } H_{i,j}^{(l)'} \neq 0 \\ 0, & \text{otherwise} \end{cases}$$

where $P_{i,j}^{(l)}$ is the element-wise activation pattern for the $j$-th feature of $i$-th node at layer $l$.

Hence, the hidden state $H^{(l)}$ at the $l$-th layer for a given sample can be written as

$$H^{(l)} = P^{(l)} \odot \left( A H^{(l-1)} \prod_{k=1}^{K} W^{\Phi_k^{(l)}} + \epsilon^{(l)} H^{(l-1)} \prod_{k=1}^{K} W^{\Phi_k^{(l)}} + \lambda H^{(l-1)} \prod_{q=1}^{Q} W^{\Psi_q^{(l)}} \right), \tag{5}$$

where $\odot$ represents element-wise multiplication. Thus, we can expand the expression of each output entry in a GNN $f(A, X)$ into a sum of scalar products, where each scalar product is the multiplication of corresponding entries in $A$, $X$, $W$, and $P$ in all layers. Then each scalar product can be written as

$$z = \mathbb{C} \cdot \left( P_{\alpha_{10},\beta_{11}}^{(1)} \cdots P_{\alpha_{L0},\beta_{L1}}^{(L)} \right) \left( P_{\alpha_{L0},\gamma_{11}}^{(c_1)} \cdots P_{\alpha_{L0},\gamma_{(M-1)1}}^{(c_{(M-1)})} \right) \cdot$$
$$\left( A_{\alpha_{L0},\alpha_{L1}}^{(L)} \cdots A_{\alpha_{10},\alpha_{11}}^{(1)} \right) X_{i,j} \left( W_{\beta_{10},\beta_{11}}^{(1)} \cdots W_{\beta_{L0},\beta_{L1}}^{(L)} \right) \left( W_{\gamma_{10},\gamma_{11}}^{(c_1)} \cdots W_{\gamma_{M0},\gamma_{M1}}^{(c_M)} \right), \tag{6}$$

where $\mathbb{C}$ is a constant, $c_k$ refers to the $k$-th layer of the classifier, $(\alpha_{l0}, \alpha_{l1}), (\beta_{l0}, \beta_{l1}), (\gamma_{l0}, \gamma_{l1})$ are (*row, column*) indices of the corresponding matrices at layer $l$. In a classifier with $M$ MLP layers, only $(M-1)$ layers adopt activation functions. Therefore, we do not have $P_{\alpha_{L0},\gamma_{M1}}^{(c_M)}$ in Equation (6). For scalar products without factors of $A$, all $A$'s are considered as constants equal to 1 in Equation (6). Since the GNN model parameters are pretrained and fixed, we only consider $A$, $X$, and all the $P$ terms as the variables in each product term, where $A$ is the dominant one as we explained in Section 2.

**Lemma 4** (**Equal Contribution variables in the GNN expansion form's scalar product**). *For a scalar product term $z$ in the expansion form of a pretrained GNN $f(\cdot)$, when the number of nodes $N$ is large, all variables in $z$ have equal contributions to the scalar product $z$.*

For the purpose of theoretical analysis, in the lemmas, we take the assumption that there are large number of nodes in the graphs. We demonstrate later in the experiments that "a large number of nodes" is not required in practical application of GOAt.

Hence, by Equation (2), we can calculate the contribution $I_\nu(z)$ of a variable $\nu$ (i.e., an entry in $A$, $X$ and $P$ matrices) to each scalar product $z$ (given by Equation (6)) by:

$$I_\nu(z) = \frac{z}{|V(z)|}, \tag{7}$$

where function $V(\cdot)$ represents the set of variables in its input, and $|V(z)|$ denotes the number of unique **variables** in $z$, e.g., $V(x^2 y) = \{x, y\}$, and $|V(x^2 y)| = 2$.

Similar to Section 3.1, an entry $f_{m,n}(A, X)$ of the output matrix $f(A, X)$ can be expressed by the sum of all the related scalar products as

$$f_{m,n}(A, X) = \sum \mathbb{C} \cdot \left( P_{\alpha_{10},\beta_{11}}^{(1)} \cdots P_{m,\beta_{L1}}^{(L)} \right) \cdot \left( P_{m,\gamma_{11}}^{(c_1)} \cdots P_{m,\gamma_{(M-1)1}}^{(c_{(M-1)})} \right) \cdot \left( A_{m,\alpha_{L1}}^{(L)} \cdots A_{\alpha_{10},\alpha_{11}}^{(1)} \right)$$
$$\cdot X_{i,j} \cdot \left( W_{\beta_{10},\beta_{11}}^{(1)} \cdots W_{\beta_{L0},\beta_{L1}}^{(L)} \right) \cdot \left( W_{\gamma_{10},\gamma_{11}}^{(c_1)} \cdots W_{\gamma_{M0},n}^{(c_M)} \right), \tag{8}$$

where summation is over all possible $(\alpha_{l0}, \alpha_{l1}), (\beta_{l0}, \beta_{l1}), (\gamma_{l0}, \gamma_{l1})$, for message passing layer $l = 1, \ldots, L$ or classifier layer $l = 1, \ldots, M$, as well as all $i, j$ indices for $X$. By summing up the contribution of each variable $\nu$ among the entries in the $A$, $X$ and $P$'s in all the scalar products in the expansion form of $f_{m,n}(\cdot)$, we can obtain the contribution of $\nu$ to $f_{m,n}(\cdot)$ as:

$$I_\nu(f_{m,n}(\cdot)) = \sum_{z \text{ in } f_{m,n}(\cdot) \text{ that contain } \nu} \frac{z}{|V(z)|}. \tag{9}$$

**Theorem 5** (**Contribution of variables in the expansion form of a pretrained GNN**). *Given Equations (7) and (9), for each variable $\nu$ (i.e., an entry in A, X and P matrices), when the number of nodes N is large, we can approximate $I_\nu(f_{m,n}(\cdot))$ by:*

$$I_\nu(f_{m,n}(\cdot)) = \sum_{z \text{ in } f_{m,n}(\cdot) \text{ that contain } \nu} \frac{O(\nu, z)}{\sum_{\rho \text{ in } z} O(\rho, z)} \cdot z, \tag{10}$$

*where $O(\nu, z)$ denotes the number of occurrences of $\nu$ among the variables of $z$.*

Recall that $|V(z)|$ stand for the number of **unique variables** in $z$. Hence the total number of occurrences of all the variables $\sum_{\rho \text{ in } z} O(\rho, z)$ is not necessarily equal to $|V(z)|$. For example, if all of $\{A^{(1)}_{\alpha_{10}, \alpha_{11}}, \ldots, A^{(L)}_{\alpha_{L0}, \alpha_{L1}}\}$ in $z$ are unique entries in $A$, then they can be considered as $L$ independent variables in the function representing $z$. If two of these occurrences of variables refer to the same entry in $A$, then there are only $(L-1)$ unique variables related to $A$.

Although Theorem 5 gives the contribution of each entry in $A$, $X$ and $P$'s, we need to further attribute $P$'s to $A$ and $X$ and allocate the contribution of each activation pattern $P^{(r)}_{a,b}$ to node features $X$ and edges $A$ by considering all non-zero features in $X_a$ of node $v_a$ and the edges within $m$ hops of node $v_a$, as these inputs may contribute to the activation pattern $P^{(r)}_{a,b}$. However, determining the exact contribution of each feature that contributes to $P^{(r)}_{a,b}$ is not straightforward due to non-linear activation. We approximately attribute all relevant features equally to $P^{(r)}_{a,b}$. That is, each input feature $\nu$ that has nonzero contribution to $P^{(r)}_{a,b}$ will share an equal contribution of $I_{P^{(r)}_{a,b}}(f_{m,n}(\cdot))/|V(P^{(r)}_{a,b})|$, where $|V(P^{(r)}_{a,b})|$ denotes the number of distinct node and edge features in $X$ and $A$ contributing to $P^{(r)}_{a,b}$, which is exactly all non-zero features in $X_a$ of node $v_a$ and the adjacency matrix entries within $r$ hops of node $v_a$. Finally, based on Equation (10), we can obtain the contribution of an input feature $\nu$ in $X$, $A$ of a graph instance to the $(m, n)$-th entry of the GNN output $f(\cdot)$ as:

$$\widehat{I}_\nu(f_{m,n}(\cdot)) = I_\nu(f_{m,n}(\cdot)) + \sum_{P^{(r)}_{a,b} \text{ in } f_{m,n}(\cdot), \text{ with } \nu \text{ in } P^{(r)}_{a,b}} \frac{I_{P^{(r)}_{a,b}}(f_{m,n}(\cdot))}{|V(P^{(r)}_{a,b})|}, \tag{11}$$

where $\nu$ is an entry in the adjacency matrix $A$ or the input feature matrix $X$, $P^{(r)}_{a,b}$ denotes an entry in all the activation patterns. Thus, we have attributed $f(\cdot)$ to each input feature of a given data instance.

Our approach meets the *completeness* axiom, which is a critical requirement in attribution methods (Sundararajan et al., 2017; Shrikumar et al., 2017; Binder et al., 2016). This axiom guarantees that the attribution scores for input features add up to the difference in the GNN's output with and without those features. Passing this sanity check implies that our approach provides a more comprehensive account of feature importance than existing methods that only rank the top features (Bajaj et al., 2021; Luo et al., 2020; Pope et al., 2019; Ying et al., 2019; Vu & Thai, 2020; Shan et al., 2021).

### 3.3 Case Study: Explaining Graph Convolutional Network (GCN)

GCNs use a simple sum in the combination step, and the adjacency matrix is normalized with the diagonal node degree matrix $D$. Hence, the hidden state of a GCN's $l$-th message-passing layer is:

$$H^{(l)} = \text{ReLU}\left(V H^{(l-1)} W^{(l)} + B^{(l)}\right), \tag{12}$$

where $V = D^{-\frac{1}{2}}(A + I)D^{-\frac{1}{2}}$ represents the normalized adjacency matrix with self-loops added. Suppose a GCN has three convolution layers and a 2-layer MLP as the classifier, then its expansion form without the activation functions $\text{ReLU}(\cdot)$ will be:

$$\begin{aligned}
f(V, X)_{\mathbf{P}} = {}& V^{(3)} V^{(2)} V^{(1)} X W^{(1)} W^{(2)} W^{(3)} W^{(c_1)} W^{(c_2)} + V^{(3)} V^{(2)} B^{(1)} W^{(2)} W^{(3)} W^{(c_1)} W^{(c_2)} \\
& + V^{(3)} B^{(2)} W^{(3)} W^{(c_1)} W^{(c_2)} + B^{(3)} W^{(c_1)} W^{(c_2)} + B^{(c_1)} W^{(c_2)} + B^{(c_2)},
\end{aligned} \tag{13}$$

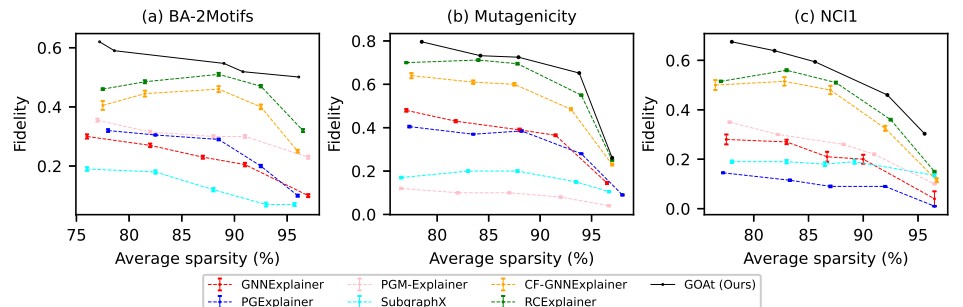

Figure 1: Fidelity performance averaged across 10 runs on the pretrained GCNs for the datasets at different levels of average sparsity.

where $V^{(l)} = V$ is the normalized adjacency matrix in the $l$-th layer's calculation. In the actual expansion form with the activation patterns, the corresponding $P^{(m)}$'s are multiplied whenever there is a $W^{(m)}$ or $B^{(m)}$ in a scalar product, excluding the last layer $W^{(c_2)}$ and $B^{(c_2)}$. For example, in the scalar products corresponding to $V^{(3)}V^{(2)}V^{(1)}XW^{(1)}W^{(2)}W^{(3)}W^{(c_1)}W^{(c_2)}$, there are eight variables consisting of four $P$'s, one $X$, and three $V$'s. By Equation (10), an adjacency entry $V_{i,j}$ itself will contribute $\frac{1}{8}$ of $p(V^{(3)}V^{(2)}_{:i}V^{(1)}_{i,j}X_j:W^{(1)}W^{(2)}W^{(3)}W^{(c_1)}W^{(c_2)}) + p(V^{(3)}_{:i}V^{(2)}_{i,j}V^{(1)}_{j:}XW^{(1)}W^{(2)}W^{(3)}W^{(c_1)}W^{(c_2)}) + p(V^{(3)}_{i,j}V^{(2)}_{j:}V^{(1)}XW^{(1)}W^{(2)}W^{(3)}W^{(c_1)}W^{(c_2)})$, where $p(\cdot)$ denotes the results with the element-wise multiplication of the corresponding activation patterns applied at the appropriate layers. After we obtain the contribution of $V_{i,j}$ itself on all the scalar products, we can follow Equation (11) to allocate the contribution of activation patterns to $V_{i,j}$.

With Equation (13), we find that when both $V$ and $X$ are set to zeros, $f(\cdot)$ remains non-zero and is:

$$f(\mathbf{0}, \mathbf{0}) = p(B^{(3)}W^{(c_1)}W^{(c_2)}) + p(B^{(c_1)}W^{(c_2)}) + B^{(c_2)}, \qquad (14)$$

where $B^{(c_2)}$ is the global bias, and the other terms have non-zero entries at the activated neurons. In other words, certain GNN neurons in the 3-rd and $c_1$-th layers may already be activated prior to any input feature being passed to the network. When we do feed input features, some of these neurons may remain activated or be toggled off. With Equation (11), we consider taking all 0's of the $X$ entries, $V$ entries and $P$ entries as the base manifold. Now, given that some of the $P$ entries in GCN are non-zero when all $X$ and $V$ set to zeros, as present in Equation (14), we will need to subtract the contribution of each features on these $P$ from the contribution values calculated by Equation (11). We let $\mathbf{P}'$ represent the activation patterns of $f(\mathbf{0}, \mathbf{0})$, then the calibrated contribution $\widehat{I}^{\text{cali}}_{V_{i,j}}(f(\cdot))$ of $V_{i,j}$ is given by:

$$\widehat{I}^{\text{cali}}_{V_{i,j}}(f(\cdot)) = \widehat{I}_{V_{i,j}}(f(V, X)) - \sum_{P'^{(r)}_{a,b} \text{ in } f(\mathbf{0},\mathbf{0}), \text{ with } V_{i,j} \text{ in } P'^{(r)}_{a,b}} \frac{I_{P'^{(r)}_{a,b}}(f(\mathbf{0},\mathbf{0}))}{|V(P^{(r)}_{a,b})|}. \qquad (15)$$

In graph classification tasks, a pooling layer such as mean-pooling is added after the convolution layers to obtain the graph representation. To determine the contribution of each input feature, we can simply apply the same pooling operation as used in the pre-trained GCN. As we mentioned in Section 2, we aim to obtain the explanations by the critical edges in this paper, since edges have more fine-grained information than nodes. Therefore, we treat the edges as variables, while considering the node features $X$ as constants similar to parameters $W$ or $B$. This setup naturally aggregates the contribution of node features onto edges. By leveraging edge attributions, we are able to effectively highlight motifs within the graph structure.

## 4 EXPERIMENTS

We evaluate the performance of explanations on three variants of GNNs, which are GCN (Kipf & Welling, 2017), GraphSAGE (Hamilton et al., 2017) and GIN (Xu et al., 2019). The experiments are conducted on six datasets including both the graph classification datasets and the node classification datasets. For graph classification task, we evaluate on a synthetic dataset, BA-2motifs (Luo et al., 2020), and two real-world datasets, Mutagenicity (Kazius et al., 2005) and NCI1 (Pires et al., 2015). For node classification task, we evaluate on three synthetic datasets (Luo et al., 2020), which are BA-shapes, BA-Community and Tree-grid. We conduct a series of experiments on the fidelity, discriminability and stability metrics to compare

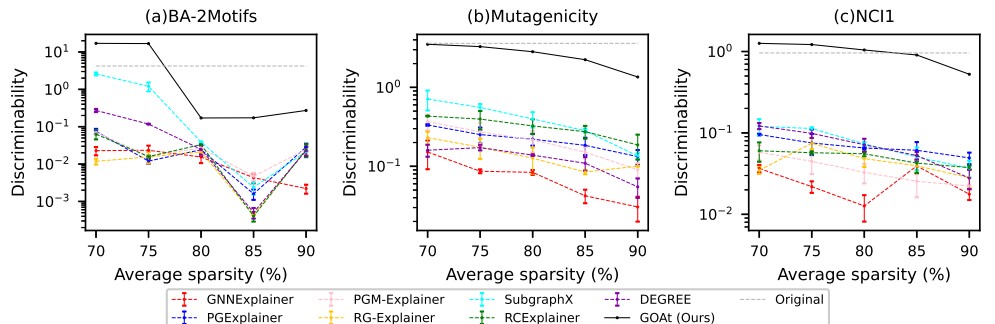

Figure 2: Discriminability performance averaged across 10 runs of the explanations produced by various GNN explainers at different levels of sparsity. "Original" refer to the performance of feeding the original data into the GNN without any modifications or explanations applied.

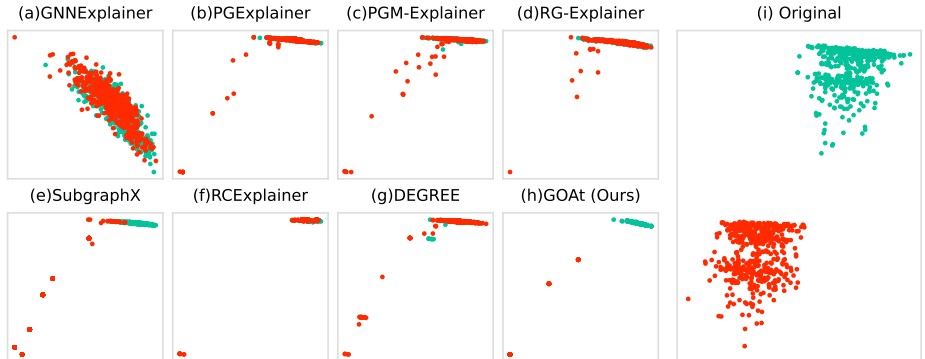

Figure 3: Visualization of explanation embeddings on the BA-2Motifs dataset. Subfigure (i) refers to the visualization of the original embeddings by directly feeding the original data into the GNN without any modifications or explanations applied.

our method with the state-of-the-art methods including GNNExplainer (Ying et al., 2019), PGExplainer (Luo et al., 2020), PGM-Explainer (Vu & Thai, 2020), SubgraphX (Yuan et al., 2021), CF-GNNExplainer (Lucic et al., 2022), RCExplainer (Bajaj et al., 2021), RG-Explainer (Shan et al., 2021) and DEGREE (Feng et al., 2022). We reran these baselines since most of them are not trained on GraphSAGE and GIN. We implemented RCExplainer as the code is not publicly available. As outlined in Section 2, we highlight edges as explanations as suggested by (Faber et al., 2021). For baselines that identify nodes or subgraphs as explanations, we adopt the evaluation setup from (Bajaj et al., 2021). As space is limited, we will only present the key results here. Fidelity results on GIN and GraphSAGE, as well as the results of node classification tasks are in Appendix G and Appendix I. Discussions on the controversial metrics such as accuracy are also moved to the Appendix I.

**Fidelity** (Pope et al., 2019; Yuan et al., 2022; Wu et al., 2022; Bajaj et al., 2021) is the decrease of predicted probability between original and new predictions after removing important edges, which are used to evaluate the faithfulness of explanations. It is defined as $fidelity(S, G) = f_y(G) - f_y(G \backslash S)$. Similar to (Yuan et al., 2022; Xie et al., 2022; Wu et al., 2022; Bajaj et al., 2021), we evaluate fidelity at different sparsity levels, where $sparsity(S, G) = 1 - \frac{|S|}{|\mathcal{E}|}$, indicating the percentage of edges that remain in $G$ after the removal of edges in $S$. Higher sparsity means fewer edges are identified as critical, which may have a smaller impact on the prediction probability. Figure 1 displays the fidelity results, with the baseline results sourced from (Bajaj et al., 2021). Our proposed approach, *GOAt*, consistently outperforms the baselines in terms of fidelity across all sparsity levels, validating its superior performance in generating accurate and reliable faithful explanations. Among the other methods, RCExplainer exhibits the highest fidelity, as it is specifically designed for fidelity optimization. Notably, unlike the other methods that require training and hyperparameter tuning, *GOAt* offers the advantage of being a training-free approach, thereby avoiding any errors across different runs.

**Discriminability.** It is also known as discrimination ability (Bau et al., 2017; Iwana et al., 2019), refers to the ability of the explanations to distinguish between the classes. We define the discriminability between two classes $c_1$ and $c_2$ as the L2 norm of the difference between the mean values of explanation embeddings $h_S^{(L)}$ of the two classes: $discriminability(c_1, c_2) = \left\| \frac{1}{N_{c_1}} \sum_{i \in c_1} h_{S_i}^{(L)} - \frac{1}{N_{c_2}} \sum_{j \in c_2} h_{S_j}^{(L)} \right\|_2$, where $N_{c_1}$ and $N_{c_2}$ refer to the number of data samples

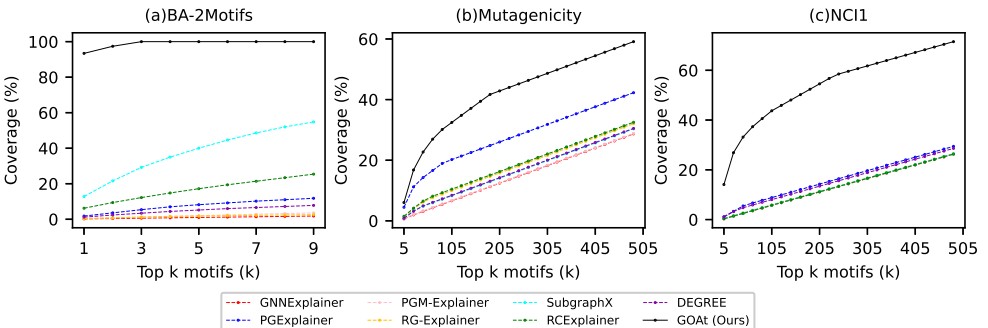

Figure 4: Coverage of the top-$k$ explanations across the datasets.

in class $c_1$ and $c_2$. The embeddings $h_S^{(L)}$ used for explanations are taken prior to the last-layer classifier, with node embeddings employed for node classification tasks and graph embeddings utilized for graph classification tasks. In this procedure, only the explanation subgraph $S$ is fed into the GNN instead of $G$. We show the discriminability across various sparsity levels on GCN, as illustrated in Figure 2. When we evaluate the discriminability, we filtered out the wrongly predicted data samples. Therefore, only when GNN makes a correct prediction, the embedding would be included in the discriminability computation. Due to the significant performance gap between the baselines and *GOAt*, a logarithmic scale is employed. Our approach consistently outperforms the baselines in terms of discriminability across all sparsity levels, demonstrating its superior ability to generate accurate and reliable class-specific explanations. Notably, at $sparsity = 0.7$, *GOAt* achieves higher discriminability than the original graphs on the BA-2Motifs and NCI1 datasets. This indicates that *GOAt* effectively reduces noise unrelated to the investigated class while producing informative class explanations. Furthermore, we present scatter plots to visualize the explanation embeddings generated by various GNN explainers. Figure 3 showcases the explanation embeddings obtained from different GNN explaining methods on the BA-2Motifs dataset, with $sparsity = 0.7$. More scatter plots on Mutagenicity and NCI1 and can be found in the Appendix **??**. The explanations generated by GNNExplainer fail to exhibit class discrimination, as all the data points are clustered together without any distinct separation. While some of the Class 1 explanations produced by PGExplainer, PGM-Explainer, RG-Explainer, RCExplainer, and DEGREE are noticeably separate from the Class 0 explanations, the majority of the data points remain closely clustered together. As for SubgraphX, most of its Class 1 explanations are isolated from the Class 0 explanations, but there is a discernible overlap between the Class 1 and Class 0 data points. In contrast, our method, *GOAt*, generates explanations that clearly and effectively distinguish between Class 0 and Class 1, with no overlapping points and a substantial separation distance, highlighting the strong discriminability of our approach.

**Stability of extracting motifs.** As we will later show in Section **??**, it is often observed that datasets contain specific class motifs. For instance, in the BA-2Motifs dataset, the Class 1 motif exhibits a "house" structure. To ensure the stability of GNN explainers in capturing the class motifs across diverse data samples, we aim for the explanation motifs to exhibit relative consistency for data samples with similar properties, rather than exhibiting significant variations. To quantify this characteristic, we introduce the *stability* metric, which measures the coverage of the top-$k$ explanations across the dataset: $stability = \frac{\sum_{S_i \in \mathcal{S}_{topk}} N_{S_i}}{N}$, where the explanations $S_i$'s are ranked by the number of data samples that each explanation can explain, i.e., $N_{S_i}$, $\mathcal{S}_{topk}$ is the set of the top-$k$ explanations based on the rank, and $N$ is total number of data samples. An ideal explainer should generate explanations that cover a larger number of data samples using fewer motifs. This characteristic is also highly desirable in global-level explainers, such as (Azzolin et al., 2023; Huang et al., 2023). We illustrate the stability of the unbiased class as the percentage converge of the top-$k$ explanations produced on GCN with $sparsity = 0.7$ in Figure 4. Our approach surpasses the baselines by a considerable margin in terms of the stability of producing explanations. Specifically, *GOAt* is capable of providing explanations for all the Class 1 data samples using only three explanations. This explains why there are only three Class 1 scatters visible in Figure 3.

**Qualitative analysis.** We present the qualitative results of our approach in Table 1, where we compare it with state-of-the-art baselines such as PGExplainer, SubgraphX, and RCExplainer. The pretrained GNN achieves a 100% accuracy on the BA-2Motifs dataset. As long as it successfully identifies one class, the remaining data samples naturally belong to the other class, leading to a perfect accuracy rate. Based on the explanations from *GOAt*, we have observed that the GNN effectively recognizes the "house" motif that is associated with Class 1. In contrast, other approaches face difficulties in consistently capturing this motif. The Class 0 motifs in the Mutagenicity dataset generated by *GOAt* represent multiple connected carbon rings. This indicates that the presence

Table 1: Qualitative results of the top motifs of each class produced by PGExplainer, SubgraphX, RCExplainer and *GOAt*. The percentages indicate the coverage of the explanations.

| | BA-2Motifs | | Mutagenicity | | NCI1 | |
|---|---|---|---|---|---|---|
| | Class0 | Class1 | Class0 | Class1 | Class0 | Class1 |
| PGExplainer | 4.8% | 1.8% | 1.2% | 1.3% | 0.1% | 0.5% |
| SubgraphX | 0.4% | 12.8% | 0.2% | 0.2% | 0.2% | 0.1% |
| RCExplainer | 6.4% | 6.2% | 0.4% | 0.5% | 0.05% | 0.1% |
| *GOAt* | 3.8%   3.4% | 93.4%   4% | 3.5%   2.2% | 2.2%   1.2% | 3.5%   1.2% | 4.3%   4.0% |

of more carbon rings in a molecule increases its likelihood of being mutagenic (Class 0), while the presence of more "C-H" or "O-H" bonds in a molecule increases its likelihood of being non-mutagenic (Class 1). Similarly, in the NCI1 dataset, *GOAt* discovers that the GNN considers a higher number of carbon rings as evidence of chemical compounds being active against non-small cell lung cancer. Other approaches, on the other hand, fail to provide clear and human-understandable explanations.

## 5   RELATED WORK

Local-level Graph Neural Network (GNN) explanation approaches have been developed to shed light on the decision-making process of GNN models at the individual data instance level. Most of them, such as GNNExplainer (Ying et al., 2019), PGExplainer (Luo et al., 2020), PGM-Explainer (Vu & Thai, 2020), GraphLime (Huang et al., 2022), RG-Explainer (Shan et al., 2021), CF-GNNExplainer (Lucic et al., 2022), RCExplainer (Bajaj et al., 2021), $CF^2$ (Tan et al., 2022), RelEx (Zhang et al., 2021) and Gem (Lin et al., 2021), TAGE (Xie et al., 2022), GStarX (Zhang et al., 2022) train a secondary model to identify crucial nodes, edges, or subgraphs that explain the behavior of pretrained GNNs for specific input samples. Zhao et al. (2023) introduces alignment objectives to improve these methods, where the embeddings of the explanation subgraph and the raw graph are aligned via anchor-based objective, distributional objective based on Gaussian mixture models, mutualinformation-based objective. However, the quality of the explanations produced by these methods is highly dependent on hyperparameter choices. Moreover, these explainers' black-box nature raises doubts about their ability to provide comprehensive explanations for GNN models. Approaches like SA (Baldassarre & Azizpour, 2019), Grad-CAM (Pope et al., 2019), GNN-LRP (Schnake et al., 2021), and DEGREE (Feng et al., 2022), which rely on gradient back-propagation, encounter the saturation problem (Shrikumar et al., 2017). As a result, these methods may generate less faithful explanations. SubgraphX (Yuan et al., 2021) combines perturbation-based techniques with pruning using Shapley values. While it can generate some high-quality subgraph explanations, its computational cost is significantly high due to the reliance on the MCTS (Monte Carlo Tree Search). Additionally, as demonstrated in our experiments in Section 4, existing methods exhibit inconsistencies on similar data samples and poor discriminability. This reinforces the need for our proposed method *GOAt*, which outperforms state-of-the-art baselines on fidelity, discriminability and stability metrics.

## 6   CONCLUSION

We propose *GOAt*, a local-level GNN explainer that overcomes the limitations of existing GNN explainers, in terms of insufficient discriminability, inconsistency on same-class data samples, and overfitting to noise. We analytically expand GNN outputs for each class into a sum of scalar products and attribute each scalar product to each input feature. Although *GOAt* shares similar limitations with some decomposition methods of requiring expert knowledge to design corresponding explaining processes for various GNNs, our extensive experiments on both synthetic and real datasets, along with qualitative analysis, demonstrate its superior explanation ability. Our method contributes to enhancing the transparency of decision-making in various fields where GNNs are widely applied.

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

## A    PROOF OF LEMMA 2

*Proof.* We prove by contradiction. Suppose Lemma 2 is false, then either of the following shall hold:

i) There exists two variables $\{X_i, X_j\} \subset \mathbf{X}$, which are not equally contribute to $g(\mathbf{X})$ at $(x_i, x_j)$ with respect to $(0,0)$;

ii) There are two variables $X_i, X_j \subset \mathbf{X}$ that cannot have $X_i = x_i$ and $X_j = 0$ simultaneously, or $X_i = 0$ and $X_j = x_j$ simultaneously.

We consider contradiction to each of the above statements separately.

i) According to Definition 1, if there exists two variables $\{X_i, X_j\} \subset \mathbf{X}$, which are not equally contribute to $g(\mathbf{X})$ at $(x_i, x_j)$ with respect to $(0,0)$, we should have at least one assignment of $\mathbf{X}$ except for $X_i$ and $X_j$ such that

$$g_{X_i=x_i, X_j=0}(\mathbf{X}\backslash\{X_i, X_j\}) \neq g_{X_i=0, X_j=x_j}(\mathbf{X}\backslash\{X_i, X_j\}).$$

However, since

$$g_{X_i=x_i, E_j=0}(\mathbf{X}\backslash\{X_i, X_j\}) = g_{X_i=0, X_j=x_j}(\mathbf{X}\backslash\{X_i, X_j\}) = 0,$$

we have reached a contradiction to the first statement.

ii) As all the variables are uncorrelated, the value assigned to one variable has no impact on how we choose values for the other variables. Therefore, it is possible to have $X_i = x_i$ and $X_j = 0$ simultaneously or to have $X_i = 0$ and $X_j = x_j$ simultaneously for arbitrary two variables $X_i$ and $X_j$. We have reached a contradiction to the second statement. $\qquad\square$

## B    PROOF OF LEMMA 4

*Proof.* To prove Lemma 4, we first prove the following Lemma.

**Lemma 6.** *For a scalar product term $z$ in the expansion form of a pretrained GNN $f(\cdot)$, when the number of nodes $N$ is large, it is feasible to have both $\nu_i = x_i$ and $\nu_j = 0$ or both $\nu_i = 0$ and $\nu_j = x_j$ for all possible pairs $\nu_i, \nu_j$ of variables in $z$, where $x_i, x_j$ indicate the presence of variables $\nu_i, \nu_j$, respectively.*

*Proof.* We first show that in the scenario of a large number of nodes $N$, an arbitrary variable $P_{c,e}$ among the variables in $z$ can take any value within its domain while keeping all other variables in $z$ fixed to certain values.

We begin with defining notations. For a scalar product term $z$ in the expansion of a pretrained GNN $f(\cdot)$, we let $\mathbf{U} = A^{(L)}_{\alpha_{L0}, \alpha_{L1}} \ldots A^{(1)}_{\alpha_{10}, \alpha_{11}}$ denotes the factors involving the adjacency matrix $A$, $\mathbf{P} = P^{(1)}_{\alpha_{10}, \beta_{11}} \ldots P^{(L)}_{\alpha_{L0}, \beta_{L1}} \cdot P^{(c_1)}_{\alpha_{L0}, \gamma_{11}} \ldots P^{(c_{(M-1)})}_{\alpha_{L0}, \gamma_{(M-1)1}}$ denotes the factors involving the activation

pattern $P$, and $\mathbb{C} = W^{(1)}_{\beta_{10},\beta_{11}} \ldots W^{(L)}_{\beta_{L0},\beta_{L1}} \cdot W^{(c_1)}_{\gamma_{10},\gamma_{11}} \ldots W^{(c_M)}_{\gamma_{M0},\gamma_{M1}}$ stands for "reduced to constant" denoting the product of the related parameters in $f(\cdot)$, such that each product term can be rewritten as $z = \mathbf{U}\mathbf{P}X_{i,j}\mathbb{C}$.

The entries in an activation pattern are determined by the hidden representation before being passed to the activation function. Similar to Equation (9), the $(c,e)$-th entry of the hidden representation at the $l$-th layer before the activation function can be expressed by the sum of all the related scalar products as

$$h^{(l)\prime}_{c,e} = \sum^{\alpha,\beta,\rho} \left[ \left( A^{(l)}_{c,\alpha_{l1}} W^{(l)}_{\beta_{l0},e} \prod_{m=1}^{l-1} P^{(m)}_{\alpha_{m0},\beta_{m1}} A^{(m)}_{\alpha_{m0},\alpha_{m1}} W^{(m)}_{\beta_{m0},\beta_{m1}} \right) X_{\rho_{m0},\rho_{m1}} \right].$$

When none of the variables in $h^{(l)\prime}_{c,e}$ are constrained, the mathematical range of $h^{(l)\prime}_{c,e}$ is $\mathbb{R}$. Let $c(h^{(l)\prime}_{c,e}, z)$ be the sum of scalar products involving the variables in $z$. If the range of $(h^{(l)\prime}_{c,e} - c(h^{(l)\prime}_{c,e}, z))$ is also $\mathbb{R}$ when none of the variables in it is constrained, then $P_{c,e}$ can take any value within its domain while keeping the variables in $z$ fixed to some certain values. In other words, if there is at least one scalar product in $(h^{(l)\prime}_{c,e} - c(h^{(l)\prime}_{c,e}, z))$, then $P_{c,e}$ can take any value within its domain while holding the variables in $z$ fixed to certain values.

The sum of scalar products involving $X_{i,j}$ is

$$c(h^{(l)\prime}_{c,e}, X_{i,j}) = \sum^{\alpha,\beta} \left[ \left( A^{(l)}_{c,\alpha_{l1}} W^{(l)}_{\beta_{l0},e} \prod_{m=1}^{l-1} P^{(m)}_{\alpha_{m0},\beta_{m1}} A^{(m)}_{\alpha_{m0},\alpha_{m1}} W^{(m)}_{\beta_{m0},\beta_{m1}} \right) X_{i,j} \right].$$

The sum of scalar products involving an arbitrary variable $A_{a,b}$ in $\mathbf{U}$ is

$$c(h^{(l)\prime}_{c,e}, A_{a,b}) = \sum^{\alpha,\beta,\rho} \left[ \left( A^{(l)}_{c,\alpha_{l1}} W^{(l)}_{\beta_{l0},e} \prod_{m=1}^{l-1} P^{(m)}_{\alpha_{m0},\beta_{m1}} A^{(m)}_{\alpha_{m0},\alpha_{m1}} W^{(m)}_{\beta_{m0},\beta_{m1}} \right) X_{\rho_{m0},\rho_{m1}} \right], \text{where}$$

at least one of $\{A^{(l)}_{c,\alpha_{l1}}, A^{(l-1)}_{\alpha_{(l-1)0},\alpha_{(l-1)1}}, \ldots, A^{(1)}_{\alpha_{10},\alpha_{11}}\}$ is $A_{a,b}$.

The sum of scalar products involving an arbitrary variable $P_{g,h}$ in $\mathbf{P}$ is

$$c(h^{(l)\prime}_{c,e}, P_{g,h}) = \sum^{\alpha,\beta,\rho} \left[ \left( A^{(l)}_{c,\alpha_{l1}} W^{(l)}_{\beta_{l0},e} \prod_{m=1}^{l-1} P^{(m)}_{\alpha_{m0},\beta_{m1}} A^{(m)}_{\alpha_{m0},\alpha_{m1}} W^{(m)}_{\beta_{m0},\beta_{m1}} \right) X_{\rho_{m0},\rho_{m1}} \right], \text{where}$$

one of $\{P^{(l-1)}_{\alpha_{(l-1)0},\beta_{(l-1)1}}, \ldots, P^{(1)}_{\alpha_{10},\beta_{11}}\}$ is $P_{g,h}$.

Then the number of scalar products in $(h^{(l)\prime}_{c,e} - c(h^{(l)\prime}_{c,e}, z))$ is

$$|h^{(l)\prime}_{c,e} - c(h^{(l)\prime}_{c,e}, z)| \geq |h^{(l)\prime}_{c,e}| - |c(h^{(l)\prime}_{c,e}, X_{i,j})| - l \cdot |c(h^{(l)\prime}_{c,e}, A_{a,b})| - (l-1) \cdot |c(h^{(l)\prime}_{c,e}, P_{g,h})|,$$

where $|h^{(l)\prime}_{c,e}|, |c(h^{(l)\prime}_{c,e}, X_{i,j})|, |c(h^{(l)\prime}_{c,e}, A_{a,b})|, |c(h^{(l)\prime}_{c,e}, P_{g,h})|$ represents the number of scalar products in $h^{(l)\prime}_{c,e}, c(h^{(l)\prime}_{c,e}, X_{i,j}), c(h^{(l)\prime}_{c,e}, A_{a,b}), c(h^{(l)\prime}_{c,e}, P_{g,h})$ respectively. Hence if we can prove $|h^{(l)\prime}_{c,e}| - |c(h^{(l)\prime}_{c,e}, X_{i,j})| - l \cdot |c(h^{(l)\prime}_{c,e}, A_{a,b})| - (l-1) \cdot |c(h^{(l)\prime}_{c,e}, P_{g,h})| \geq 1$, then we will also have $|h^{(l)\prime}_{c,e} - c(h^{(l)\prime}_{c,e}, z)| \geq 1$ proved. That is, we should prove

$$\frac{|h^{(l)\prime}_{c,e}|}{|h^{(l)\prime}_{c,e}|} - \frac{|c(h^{(l)\prime}_{c,e}, X_{i,j})|}{|h^{(l)\prime}_{c,e}|} - l \cdot \frac{|c(h^{(l)\prime}_{c,e}, A_{a,b})|}{|h^{(l)\prime}_{c,e}|} - (l-1) \cdot \frac{|c(h^{(l)\prime}_{c,e}, P_{g,h})|}{|h^{(l)\prime}_{c,e}|} \geq \frac{1}{|h^{(l)\prime}_{c,e}|}.$$

Equivalently, we should prove

$$\frac{|c(h^{(l)\prime}_{c,e}, X_{i,j})|}{|h^{(l)\prime}_{c,e}|} + l \cdot \frac{|c(h^{(l)\prime}_{c,e}, A_{a,b})|}{|h^{(l)\prime}_{c,e}|} + (l-1) \cdot \frac{|c(h^{(l)\prime}_{c,e}, P_{g,h})|}{|h^{(l)\prime}_{c,e}|} \leq 1 - \frac{1}{|h^{(l)\prime}_{c,e}|}.$$

Note that the first term

$$\frac{|c(h^{(l)\prime}_{c,e}, X_{i,j})|}{|h^{(l)\prime}_{c,e}|} = \frac{1}{Nd}.$$

Since $d \geq 1$ is the feature dimension of $X$, we have $\lim_{N \to \infty} \frac{|c(h_{c,e}^{(l)'}, X_{i,j})|}{|h_{c,e}^{(l)'}|} = 0$.

Now consider the second term $l \cdot \frac{|c(h_{c,e}^{(l)'}, A_{a,b})|}{|h_{c,e}^{(l)'}|}$:

$$l \cdot \frac{|c(h_{c,e}^{(l)'}, A_{a,b})|}{|h_{c,e}^{(l)'}|} = l \cdot \frac{\binom{l}{1} N^{(l-1)} + \binom{l}{2} N^{(l-2)} + \cdots + \binom{l}{l} N^0}{N^{(l+1)}}$$

$$= l \cdot \left( \frac{\binom{l}{1}}{N^2} + \frac{\binom{l}{2}}{N^3} + \cdots + \frac{\binom{l}{l}}{N^{(l+1)}} \right)$$

$$= l \cdot \left( \frac{l!}{1!(l-1)! \cdot N^2} + \frac{l!}{2!(l-2)! \cdot N^3} + \cdots + \frac{l!}{l!(l-l)! \cdot N^{(l+1)}} \right)$$

$$= \left( \frac{l}{1!N} \cdot \frac{l}{N} \right) + \left( \frac{l}{2!N} \cdot \frac{l}{N} \cdot \frac{l-1}{N} \right) + \cdots + \left( \frac{l}{l!N} \cdot \frac{l}{N} \cdot \frac{l-1}{N} \cdots \frac{1}{N} \right).$$

Since $N \gg l$, we have $\lim_{N \to \infty} \frac{l}{N} = 0$. Also, because $\frac{l}{N} > \frac{l-1}{N} > \cdots > \frac{1}{N} > \frac{1}{1!N} > \cdots > \frac{1}{l!N}$, we then have $\lim_{N \to \infty} l \cdot \frac{|c(h_{c,e}^{(l)'}, A_{a,b})|}{|h_{c,e}^{(l)'}|} = 0$.

Now we consider the third term $(l-1) \cdot \frac{|c(h_{c,e}^{(l)'}, P_{g,h})|}{|h_{c,e}^{(l)'}|}$:

$$(l-1) \cdot \frac{|c(h_{c,e}^{(l)'}, P_{g,h})|}{|h_{c,e}^{(l)'}|} = (l-1) \cdot \frac{1}{N^l d}$$

Since $N \gg l$ and $d \geq 1$, we have $\lim_{N \to \infty} (l-1) \cdot \frac{|c(h_{c,e}^{(l)'}, P_{g,h})|}{|h_{c,e}^{(l)'}|} = 0$.

For the terms on the right hand side of the inequality, since $\frac{1}{|h_{c,e}^{(l)'}|} \leq \frac{|c(h_{c,e}^{(l)'}, X_{i,j})|}{|h_{c,e}^{(l)'}|}$, we have $\lim_{N \to \infty} \frac{1}{|h_{c,e}^{(l)'}|} = 0$. Hence $\lim_{N \to \infty} 1 - \frac{1}{|h_{c,e}^{(l)'}|} = 1$.

Since $0 < 1$, we have prove that when $N$ is large, $\frac{|c(h_{c,e}^{(l)'}, X_{i,j})|}{|h_{c,e}^{(l)'}|} + l \cdot \frac{|c(h_{c,e}^{(l)'}, A_{a,b})|}{|h_{c,e}^{(l)'}|} + (l-1) \cdot \frac{|c(h_{c,e}^{(l)'}, P_{g,h})|}{|h_{c,e}^{(l)'}|} \leq 1 - \frac{1}{|h_{c,e}^{(l)'}|}$. Therefore, in scenarios with a large number of nodes $N$, an arbitrary variable $P_{c,e}$ in $\mathbf{P}$ can take any value within its domain while keeping all other variables in $z$ fixed to certain values.

If the data is properly preprocessed, a feature $X_{i,j}$ and the unique entries in $A$ should be uncorrelated with each other. Also, from the above proof we can conclude that when $N$ is large, any arbitrary variables in $z$ can be freely set to "*absence*" or "*present*" without affecting other variables. That is, in scenarios with a large number of nodes $N$, it is always feasible to hold

- both $X_{i,j} = 0$ and $A_{a,b} = A_{a,b}$, as well as both $X_{i,j} = x_{i,j}$ and $A_{a,b} = 0$;
- both $A_{k,n} = 0$ and $A_{a,b} = A_{a,b}$, as well as both $A_{k,n} = A_{k,n}$ and $A_{a,b} = 0$;
- both $X_{i,j} = 0$ and $P_{c,e} = p_{c,e}$, as well as both $X_{i,j} = x_{i,j}$ and $P_{c,e} = 0$;
- both $A_{a,b} = 0$ and $P_{c,e} = p_{c,e}$, as well as both $A_{a,b} = A_{a,b}$ and $P_{c,e} = 0$;
- both $P_{c,e} = 0$ and $P_{g,h} = p_{g,h}$, as well as $P_{c,e} = p_{c,e}$ and $P_{g,h} = 0$,

without affecting other variables in $z$, where $X_{i,j}, A_{a,b}, A_{k,n}, P_{c,e}, P_{g,h}$ refers to the variables in $z$. Hence we have proved Lemma 6. $\qquad \square$

Next, we prove by contradiction that for all possible variable pairs $(\nu_i, \nu_j)$ among the unique variables in $z$, we have $(\nu_i, \nu_j)$ contribute equally to z at $(x_i, x_j)$ with respect to $(0, 0)$, where $\nu_i = x_i, \nu_j = x_j$ means the "*presence*" of the variables.

Assume there exists two variables $(\nu_i, \nu_j)$ in $V(z)$ that are not equally contribute to $z$ at $(x_i, x_j)$ with respect to $(0, 0)$. Then by Definition 1, we should have one assignment of other variables in $z$, such that

$$z_{\nu_i=x_i, \nu_j=0}(V(z)\backslash\{\nu_i, \nu_j\}) \neq z_{\nu_i=0, \nu_j=x_j}(V(z)\backslash\{\nu_i, \nu_j\}).$$

However, since

$$z_{\nu_i=x_i, \nu_j=0}(V(z)\backslash\{\nu_i, \nu_j\}) = z_{\nu_i=0, \nu_j=x_j}(V(z)\backslash\{\nu_i, \nu_j\}) = 0,$$

we have reached a contradiction. Hence we have proved Lemma 4. $\qquad \square$

## C    PROOF OF THEOREM 5

*Proof.* If the total number of unique variables in a scalar product equals to the total number of occurences of all the unique variables, i.e., if $|V(z)| = \sum_{\rho \text{ in } z} O(\rho, z)$, we will have $I_\nu(z) = \frac{z}{|V(z)|} = \frac{O(\nu, z) \cdot z}{\sum_{\rho \text{ in } z} O(\rho, z)}$. This is because when $|V(z)| = \sum_{\rho \text{ in } z} O(\rho, z)$, all the occurrences of variables are unique variables, and we have $O(\nu, z) = 1$. Consider $I_\nu(f_{m,n}(\cdot))$ as the sum of two components, which are the contribution $I_\nu(f_{m,n}^{V=O}(\cdot))$ of $\nu$ to the scalar products where $|V(z)| = \sum_{\rho \text{ in } z} O(\rho, z)$ holds, and the contribution $I_\nu(f_{m,n}^{V\neq O}(\cdot))$ of $\nu$ to the scalar products where $|V(z)| = \sum_{\rho \text{ in } z} O(\rho, z)$ does not hold. That is,

$$I_\nu(f_{m,n}(\cdot)) = I_\nu(f_{m,n}^{V=O}(\cdot)) + I_\nu(f_{m,n}^{V\neq O}(\cdot)).$$

We are not able to have multiple occurrences of $X$ or $P$ in a scalar product, but only able to have multiple occurrences of $A$. Considering the scalar products are bounded by a value of $c$, we have

$$\frac{I_\nu(f_{m,n}^{V\neq O}(\cdot))}{I_\nu(f_{m,n}(\cdot))} \leq \frac{c \cdot \left[\binom{L}{2}N^{L-1} + \cdots + \binom{L}{L}N\right]}{c \cdot N^{L+1}}$$

$$= \frac{L!N^{L-1}}{2!(L-2)!N^{L+1}} + \cdots + \frac{L!N}{(L)!0!N^{L+1}}$$

$$= \frac{L(L-1)}{2!N^2} + \cdots + \frac{1}{N^L}$$

$$\leq \frac{L^2}{N^2} + \cdots + \frac{L^2}{N^2},$$

Since $N \gg L$, we have

$$\lim_{N\to\infty} \frac{I_\nu(f_{m,n}^{V\neq O}(\cdot))}{I_\nu(f_{m,n}(\cdot))} = 0.$$

Therefore, when $N$ is large, $I_\nu(f_{m,n}(\cdot)) = I_\nu(f_{m,n}^{V=O}(\cdot))$. Hence, by Equation (10), we have proved that when $N$ is large, $I_\nu(f_{m,n}(\cdot)) = \sum_{z \text{ in } f_{m,n}(\cdot) \text{ that contain } \nu} \frac{O(\nu, z)}{\sum_{\rho \text{ in } z} O(\rho, z)} \cdot z$. $\qquad \square$

## D    CASE STUDY: EXPLAINING GRAPH ISOMORPHISM NETWORK (GIN)

GIN adopts weighted sum at the COMBINE step, hence the hidden state of a GIN's $l$-th layer is:

$$H^{(l)} = \Phi^{(l)}\left(\hat{A}H^{(l-1)} + \epsilon^{(l)}H^{(l-1)}\right), \tag{16}$$

where $\hat{A} = A + I$ refers to the adjacency matrix with the self-loops, $\epsilon^{(l)}$ is a trainable scalar parameter. If $\Phi^{(l)}(\cdot)$ is a 2-layer MLP, expanding $\Phi^{(l)}$, we have

$$H^{(l)} = \text{ReLU}\left(\text{ReLU}\left(\hat{A}H^{(l-1)}W^{\Phi_1^{(l)}} + \epsilon^{(l)}H^{(l-1)}W^{\Phi_1^{(l)}} + B^{\Phi_1^{(l)}}\right)W^{\Phi_2^{(l)}} + B^{\Phi_2^{(l)}}\right). \tag{17}$$

Suppose a GIN $f(\hat{A}, X)$ has three message-passing layers and a 2-layer MLP as the classifier, then its expansion form without the activation functions ReLU($\cdot$) will be

$$
\begin{aligned}
f(\hat{A}, X)_{\mathbb{P}} = \ & X\epsilon^{(3)}\epsilon^{(2)}\epsilon^{(1)}W^{\Phi_1^{(1)}}W^{\Phi_2^{(1)}}W^{\Phi_1^{(2)}}W^{\Phi_2^{(2)}}W^{\Phi_1^{(3)}}W^{\Phi_2^{(3)}}W^{(c_1)}W^{(c_2)} \\
& + \hat{A}^{(1)}X\epsilon^{(3)}\epsilon^{(2)}W^{\Phi_1^{(1)}}W^{\Phi_2^{(1)}}W^{\Phi_1^{(2)}}W^{\Phi_2^{(2)}}W^{\Phi_1^{(3)}}W^{\Phi_2^{(3)}}W^{(c_1)}W^{(c_2)} \\
& + \hat{A}^{(2)}X\epsilon^{(3)}\epsilon^{(1)}W^{\Phi_1^{(1)}}W^{\Phi_2^{(1)}}W^{\Phi_1^{(2)}}W^{\Phi_2^{(2)}}W^{\Phi_1^{(3)}}W^{\Phi_2^{(3)}}W^{(c_1)}W^{(c_2)} \\
& + \hat{A}^{(3)}X\epsilon^{(2)}\epsilon^{(1)}W^{\Phi_1^{(1)}}W^{\Phi_2^{(1)}}W^{\Phi_1^{(2)}}W^{\Phi_2^{(2)}}W^{\Phi_1^{(3)}}W^{\Phi_2^{(3)}}W^{(c_1)}W^{(c_2)} \\
& + \hat{A}^{(2)}\hat{A}^{(1)}X\epsilon^{(3)}W^{\Phi_1^{(1)}}W^{\Phi_2^{(1)}}W^{\Phi_1^{(2)}}W^{\Phi_2^{(2)}}W^{\Phi_1^{(3)}}W^{\Phi_2^{(3)}}W^{(c_1)}W^{(c_2)} \\
& + \hat{A}^{(3)}\hat{A}^{(1)}X\epsilon^{(2)}W^{\Phi_1^{(1)}}W^{\Phi_2^{(1)}}W^{\Phi_1^{(2)}}W^{\Phi_2^{(2)}}W^{\Phi_1^{(3)}}W^{\Phi_2^{(3)}}W^{(c_1)}W^{(c_2)} \\
& + \hat{A}^{(3)}\hat{A}^{(2)}X\epsilon^{(1)}W^{\Phi_1^{(1)}}W^{\Phi_2^{(1)}}W^{\Phi_1^{(2)}}W^{\Phi_2^{(2)}}W^{\Phi_1^{(3)}}W^{\Phi_2^{(3)}}W^{(c_1)}W^{(c_2)} \\
& + \hat{A}^{(3)}\hat{A}^{(2)}\hat{A}^{(1)}XW^{\Phi_1^{(1)}}W^{\Phi_2^{(1)}}W^{\Phi_1^{(2)}}W^{\Phi_2^{(2)}}W^{\Phi_1^{(3)}}W^{\Phi_2^{(3)}}W^{(c_1)}W^{(c_2)} \\
& + \epsilon^{(3)}\epsilon^{(2)}B^{\Phi_1^{(1)}}W^{\Phi_2^{(1)}}W^{\Phi_1^{(2)}}W^{\Phi_2^{(2)}}W^{\Phi_1^{(3)}}W^{\Phi_2^{(3)}}W^{(c_1)}W^{(c_2)} \\
& + \hat{A}^{(2)}\epsilon^{(3)}B^{\Phi_1^{(1)}}W^{\Phi_2^{(1)}}W^{\Phi_1^{(2)}}W^{\Phi_2^{(2)}}W^{\Phi_1^{(3)}}W^{\Phi_2^{(3)}}W^{(c_1)}W^{(c_2)} \\
& + \hat{A}^{(3)}\epsilon^{(2)}B^{\Phi_1^{(1)}}W^{\Phi_2^{(1)}}W^{\Phi_1^{(2)}}W^{\Phi_2^{(2)}}W^{\Phi_1^{(3)}}W^{\Phi_2^{(3)}}W^{(c_1)}W^{(c_2)} \\
& + \hat{A}^{(3)}\hat{A}^{(2)}B^{\Phi_1^{(1)}}W^{\Phi_2^{(1)}}W^{\Phi_1^{(2)}}W^{\Phi_2^{(2)}}W^{\Phi_1^{(3)}}W^{\Phi_2^{(3)}}W^{(c_1)}W^{(c_2)} \\
& + \epsilon^{(3)}\epsilon^{(2)}B^{\Phi_2^{(1)}}W^{\Phi_1^{(2)}}W^{\Phi_2^{(2)}}W^{\Phi_1^{(3)}}W^{\Phi_2^{(3)}}W^{(c_1)}W^{(c_2)} \\
& + \hat{A}^{(2)}\epsilon^{(3)}B^{\Phi_2^{(1)}}W^{\Phi_1^{(2)}}W^{\Phi_2^{(2)}}W^{\Phi_1^{(3)}}W^{\Phi_2^{(3)}}W^{(c_1)}W^{(c_2)} \\
& + \hat{A}^{(3)}\epsilon^{(2)}B^{\Phi_2^{(1)}}W^{\Phi_1^{(2)}}W^{\Phi_2^{(2)}}W^{\Phi_1^{(3)}}W^{\Phi_2^{(3)}}W^{(c_1)}W^{(c_2)} \\
& + \hat{A}^{(3)}\hat{A}^{(2)}B^{\Phi_2^{(1)}}W^{\Phi_1^{(2)}}W^{\Phi_2^{(2)}}W^{\Phi_1^{(3)}}W^{\Phi_2^{(3)}}W^{(c_1)}W^{(c_2)} \\
& + \hat{A}^{(3)}B^{\Phi_1^{(2)}}W^{\Phi_2^{(2)}}W^{\Phi_1^{(3)}}W^{\Phi_2^{(3)}}W^{(c_1)}W^{(c_2)} \\
& + \epsilon^{(3)}B^{\Phi_1^{(2)}}W^{\Phi_2^{(2)}}W^{\Phi_1^{(3)}}W^{\Phi_2^{(3)}}W^{(c_1)}W^{(c_2)} \\
& + \hat{A}^{(3)}B^{\Phi_2^{(2)}}W^{\Phi_1^{(3)}}W^{\Phi_2^{(3)}}W^{(c_1)}W^{(c_2)} + \epsilon^{(3)}B^{\Phi_2^{(2)}}W^{\Phi_1^{(3)}}W^{\Phi_2^{(3)}}W^{(c_1)}W^{(c_2)} \\
& + B^{\Phi_1^{(3)}}W^{\Phi_2^{(3)}}W^{(c_1)}W^{(c_2)} + B^{\Phi_2^{(3)}}W^{(c_1)}W^{(c_2)} + B^{(c_1)}W^{(c_2)} + B^{(c_2)}.
\end{aligned}
\tag{18}
$$

Then similar to the case study on GCN and GraphSAGE, the activation patterns are multiplied to each of the scalar products. Although Equation (18) may appear complex, we can observe a pattern that when $\hat{A}^{(l)}$ is present in a product term, the corresponding $\epsilon^{(l)}$ is not. This observation allows us to simplify the expression by using for loops to cover all the product terms. The code of explaining GIN on the graph classification task is in the package of Supplementary Material.

## E  CASE STUDY: EXPLAINING GRAPHSAGE (SAmple and aggreGatE)

GraphSAGE adopts concatenation at the COMBINE step, hence the hidden state of a GraphSAGE's $l$-th layer is

$$
H^{(l)} = \text{ReLU}\left(AH^{(l-1)}W^{(l)\phi} + H^{(l-1)}W^{(l)\psi} + B^{(l)}\right),
\tag{19}
$$

where $W^{(l)\phi}$ and $W^{(l)\psi}$ represents the trainable parameters for concatenating the node information and its neighborhood information. Suppose a GraphSAGE network $f(A, X)$ has three message-passing layers and a 2-layer MLP as the classifier, then its expansion form without the activation

functions ReLU($\cdot$) will be

$$
\begin{aligned}
f(A, X)_{\mathbf{P}} = {} & X W^{(1)\psi} W^{(2)\psi} W^{(3)\psi} W^{(c_1)} W^{(c_2)} + A^{(1)} X W^{(1)\phi} W^{(2)\psi} W^{(3)\psi} W^{(c_1)} W^{(c_2)} \\
& + A^{(2)} X W^{(1)\psi} W^{(2)\phi} W^{(3)\psi} W^{(c_1)} W^{(c_2)} + A^{(3)} X W^{(1)\psi} W^{(2)\psi} W^{(3)\phi} W^{(c_1)} W^{(c_2)} \\
& + A^{(2)} A^{(1)} X W^{(1)\phi} W^{(2)\phi} W^{(3)\psi} W^{(c_1)} W^{(c_2)} \\
& + A^{(3)} A^{(1)} X W^{(1)\phi} W^{(2)\psi} W^{(3)\phi} W^{(c_1)} W^{(c_2)} \\
& + A^{(3)} A^{(2)} X W^{(1)\psi} W^{(2)\phi} W^{(3)\phi} W^{(c_1)} W^{(c_2)} \\
& + A^{(3)} A^{(2)} A^{(1)} X W^{(1)\phi} W^{(2)\phi} W^{(3)\phi} W^{(c_1)} W^{(c_2)} \\
& + A^{(2)} B^{(1)} W^{(2)\phi} W^{(3)\psi} W^{(c_1)} W^{(c_2)} + A^{(3)} B^{(1)} W^{(2)\psi} W^{(3)\phi} W^{(c_1)} W^{(c_2)} \\
& + A^{(3)} A^{(2)} B^{(1)} W^{(2)\phi} W^{(3)\phi} W^{(c_1)} W^{(c_2)} + B^{(1)} W^{(2)\psi} W^{(3)\psi} W^{(c_1)} W^{(c_2)} \\
& + A^{(3)} B^{(2)} W^{(3)\phi} W^{(c_1)} W^{(c_2)} + B^{(2)} W^{(3)\psi} W^{(c_1)} W^{(c_2)} \\
& + B^{(3)} W^{(c_1)} W^{(c_2)} + B^{(c_1)} W^{(c_2)} + B^{(c_2)}.
\end{aligned}
\tag{20}
$$

Then all the other steps will be identical to the case study of GCN. The code of explaining Graph-SAGE on the graph classification task is in the package of Supplementary Material.

## F  HANDLING BATCH NORMALIZATION LAYER

In certain cases, Batch Normalization (BN) may be applied between the message-passing layers. In this section, we will elaborate on how BN layer is handled to provide explantions with *GOAt*. The formula of BN is

$$
y = \frac{x - \mu}{\sqrt{\delta + \varepsilon}} \cdot W + B,
\tag{21}
$$

where $\mu$ is the running mean, $\delta$ is the running variance, $\varepsilon$ is a prefixed small value, $W$, $B$ are learnable parameters. During the evaluation mode of a pretrained GNN, $\mu, \delta, \varepsilon, W$ and $B$ are fixed. As a result, we can treat the Batch Normalization (BN) layer as a linear mapping $y = x W^{(\mathrm{BN})} + B^{(\mathrm{BN})}$ while obtaining GNN explanations with *GOAt*, where

$$
W^{(\mathrm{BN})} = \frac{W}{\sqrt{\delta + \varepsilon}}, \; B^{(\mathrm{BN})} = \frac{-\mu \cdot W}{\sqrt{\delta + \varepsilon}} + B.
\tag{22}
$$

## G  FIDELITY RESULTS OF EXPLAINING GRAPHSAGE AND GIN

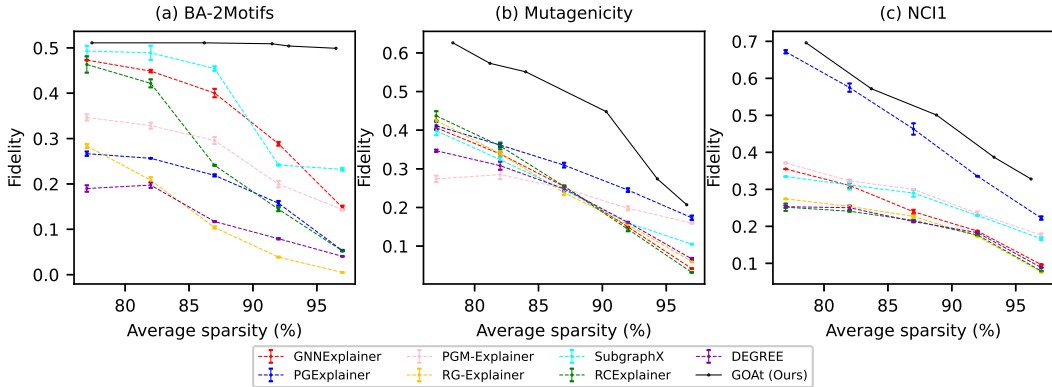

Figure 5: Fidelity performance averaged across 10 runs on the pretrained GraphSAGE for the datasets at different levels of average sparsity.

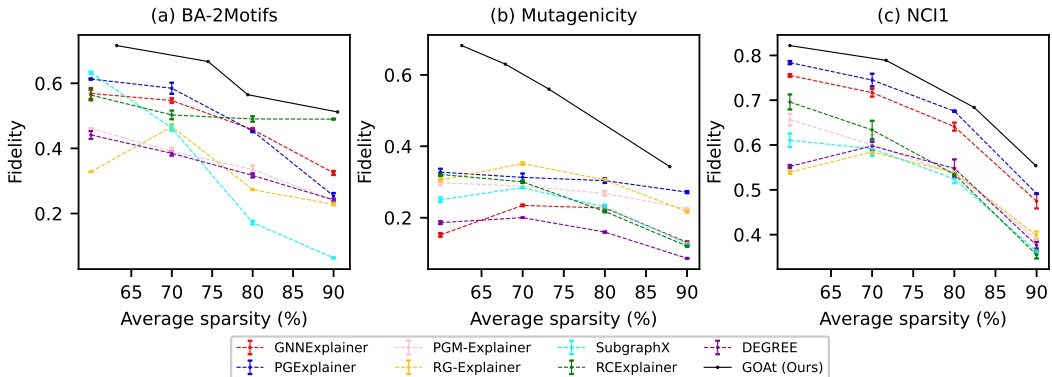

Figure 6: Fidelity performance averaged across 10 runs on the pretrained GIN for the datasets at different levels of average sparsity.

## H    STATISTICS AND IMPLEMENTATION DETAILS

The GNNs are trained using the following data splits: 80% for the training set, 10% for the validation set, and 10% for the testing set. All experiments are conducted on an Intel® Core™ i7-10700 Processor and NVIDIA GeForce RTX 3090 Graphics Card. The GNN architectures consist of 3 message-passing layers and a 2-layer classifier. The hidden dimension is set to 32 for BA-2Motifs, BA-Shapes, BA-Community, Tree-grid, and 64 for Mutagenicity and NCI1. The code is available in the Supplementary Material, provided alongside this Appendix file.

Table 2: Statistics of the datasets used and the classification accuracy of the trained GNNs.

|  |  | BA-Shapes | BA-Community | Tree-Grid | BA-2Motifs | Mutagenicity | NCI1 |
|---|---|---|---|---|---|---|---|
| # Graphs |  | 1 | 1 | 1 | 1,000 | 4,337 | 4,110 |
| # Nodes (avg) |  | 700 | 1,400 | 1,231 | 25 | 30.32 | 29.87 |
| # Edges (avg) |  | 4,110 | 8,920 | 3,410 | 25.48 | 30.77 | 32.30 |
| # Classes |  | 4 | 8 | 2 | 2 | 2 | 2 |
| Test ACC | GCN | 0.97 | 0.91 | 0.97 | 1.00 | 0.82 | 0.81 |
|  | GraphSAGE | - | - | - | 1.00 | 0.80 | 0.80 |
|  | GIN | - | - | - | 1.00 | 0.89 | 0.83 |

## I    EXPLANATIONS OF NODE CLASSIFICATION

We utilize scatter plots to visually depict the explanation embeddings produced by GNNExplainer, PGExplainer, and GOAt, and compare them with the node embeddings in the original graphs. In the generated figures, we set the value of $topk$ to be 6, 7, and 14 for the BA-Shapes, BA-Community, and Tree-Grid datasets, respectively. In the case of BA-Shapes and BA-Community, we only plot the nodes within the house-shape motif, as the other nodes are located far away and may not be easily discernible in terms of explanation performance.

As presented in Figure 9, the majority of the explanations on the Tree-Grid dataset generated by GNNExplainer are closely clustered together, and *GOAt* has fewer overlapped data points than PG-Explainer. As illustrated in Figure 7 and Figure 8, the explanations generated by GNNExplainer and PGExplainer fail to exhibit class discrimination on BA-Shapes and BA-Communicty datasets, as all the data points are clustered together without any distinct separation. In contrast, our method, *GOAt*, generates explanations that clearly and effectively distinguish between classes, with fewer overlapping points and substantial separation distances, highlighting the strong discriminability of our approach on the node classification task.

**Discussion on the AUC/Accuracy metrics.** Many existing GNN explanation approaches are evaluated using metrics such as AUC or Accuracy. These metrics compare the explanations generated by the explainers with "ground-truth" explanations that are predetermined by humans. Ground-truth explanations refer to the underlying evidence that leads to the correct label, rather than the prediction

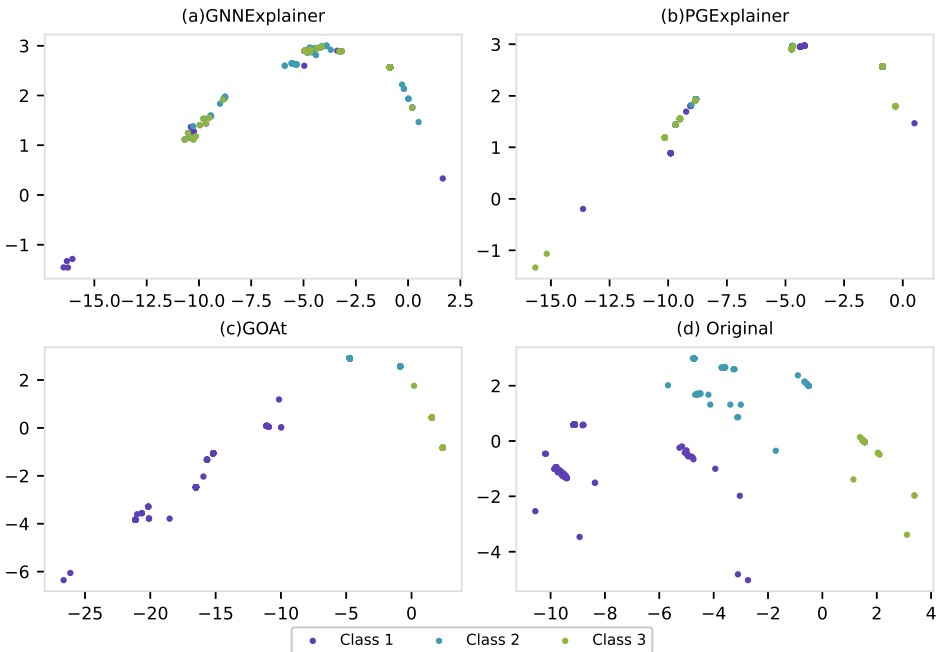

Figure 7: Visualization of explanation embeddings on the BA-Shapes dataset. Subfigure (d) refers to the visualization of the original embeddings by directly feeding the original data into the GNN without any modifications or explanations applied.

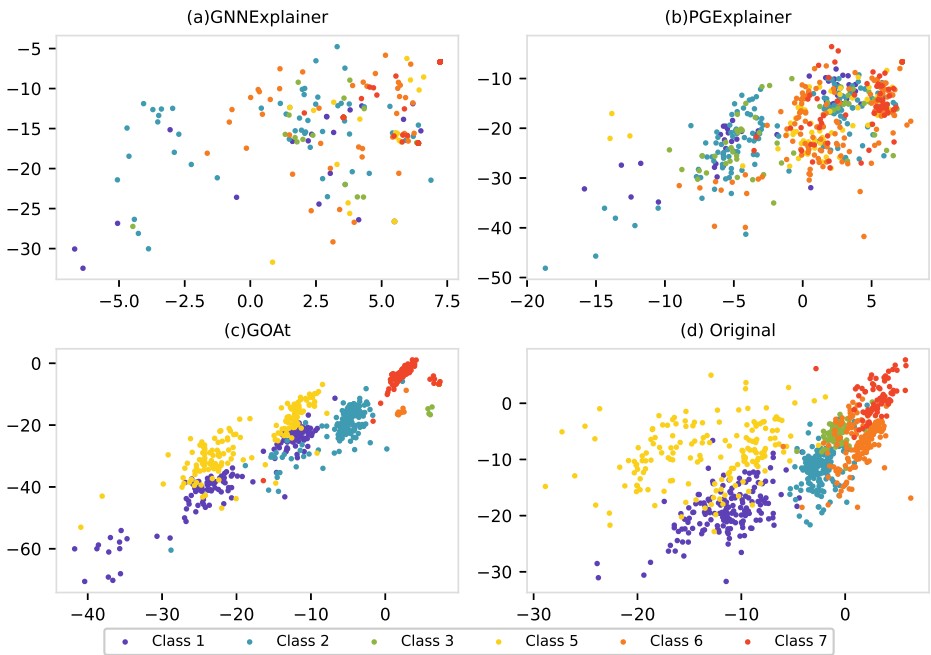

Figure 8: Visualization of explanation embeddings on the BA-Community dataset. Subfigure (d) refers to the visualization of the original embeddings by directly feeding the original data into the GNN without any modifications or explanations applied.

label itself. However, as highlighted by (Faber et al., 2021) there can often be a mismatch between the ground truth and the GNN. To avoid any potential misunderstandings, we have chosen to directly present scatter plots of the explanations generated by different explainers.

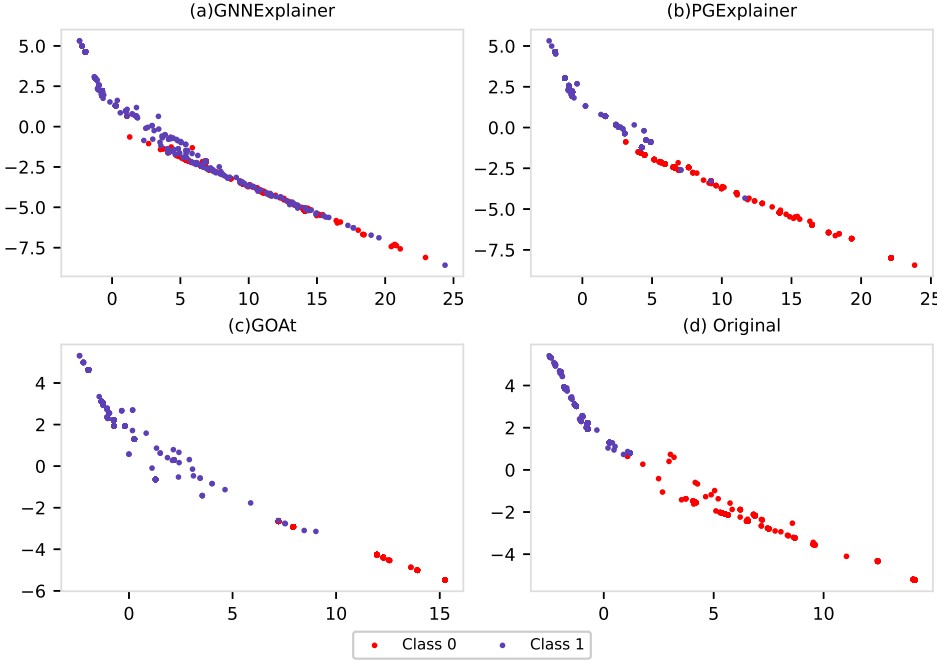

Figure 9: Visualization of explanation embeddings on the Tree-Grid dataset. Subfigure (d) refers to the visualization of the original embeddings by directly feeding the original data into the GNN without any modifications or explanations applied.

## J TIME ANALYSIS

Table 3 presents the average running time per data sample over the six datasets used in our experiments on GCNs. We compared with the outstanding baselines, which are the perturbation-based method GNNExplainer (Ying et al., 2019), decomposition-based method DEGREE (Feng et al., 2022), and the search-based method SubgraphX (Yuan et al., 2021). Our approach is much faster than SubgraphX and GNNExplainer, and runs slightly faster than DEGREE while providing more faithful, discriminative and stable explanations as demonstrated in Section 4, which showcases the significance of our method.

Table 3: Average running time per data sample.

|              | BA-Shapes   | BA-Community | Tree-Grid   | BA-2Motifs  | Mutagenicity  | NCI1         |
|--------------|-------------|--------------|-------------|-------------|---------------|--------------|
| GNNExplainer | 0.65±0.05s  | 0.78±0.05s   | 0.72±0.06s  | 1.16±0.10s  | 1.43±0.03s    | 1.44±0.09s   |
| DEGREE       | 0.44±0.20s  | 1.02±0.35s   | 0.37±0.06s  | 0.58±0.11s  | 0.83±0.45s    | 0.94±0.34s   |
| SubgraphX    | 81.8±34.9s  | 138.8±38.3s  | 56.7±12.1s  | 95.4±11.7s  | 419.8±655.6s  | 541.6±357.0s |
| GOAt (ours)  | 0.36±0.00s  | 0.50±0.00s   | 0.05±0.00s  | 0.46±0.00s  | 0.60±0.22s    | 0.83±0.39s   |

## K ILLUSTRATIVE EXAMPLE OF GOAT

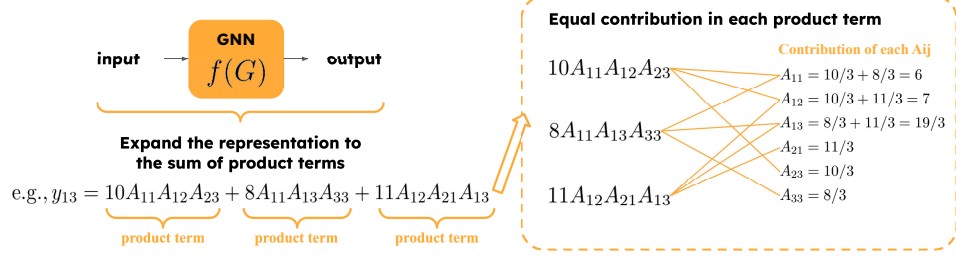

Figure 10: Illustrative example of GOAt.

