# OpenReview forum: "GOAt: Explaining Graph Neural Networks via Graph Output Attribution"
_ICLR.cc/2024/Conference — ICLR 2024 poster_

### Official Review · Reviewer_nBUo · 2023-10-27

**Soundness:** 3 good
**Presentation:** 2 fair
**Contribution:** 3 good
**Rating:** 6
**Confidence:** 5

**Summary:**

In this paper, the authors introduce  a new method for interpreting GNNs. Specifically, this method provides transparent explanations by attributing graph outputs to input features. It employs an analytical approach that expands the GNN into a sum of scalar products, which allows for efficient calculation of feature contributions. Empirical results on three well-known datasets show the effectiveness of the proposed method.

**Strengths:**

1.Unlike previous methods that either rely on back-propagation with gradients or train complex black-box models, this method uses an analytical method based on the sum-product structure of the GNN’s forward pass.
2. The proposed method demonstrates promising performance on synthetic and real-world data. These experiments demonstrate that GOAt outperforms existing state-of-the-art methods in several key metrics, such as fidelity, discriminability, and stability.
3. The paper seems to be well-organized in its presentation.

**Weaknesses:**

1. It isunclear what limitations of previous methods the author's method can overcome, and the reasons for overcoming these limitations. It would be good to clarify these two points.
2. Please explain why it makes sense to define each edge as an Equal Contribution.
3. The latest methods mentioned in the Related Work: GLGExplainer and GCFExplainer, do not appear in the experiments as baselines.
4 .The experimental datasets could be more diverse, such as including the BBBP, MNIST, and other datasets, to make the model more convincing.
5. The reference to 'Equation(??)' in the paragraph below Equation 5 is unclear.

**Questions:**

Please refer to Weaknesses 1-5.

---

> ### Author Response · Authors · 2023-11-23
>
> **W1**: First, as discussed in Sections 1 and 5, most existing instance-level methods train a secondary model to identify the critical graph structures, resulting in black-boxed explanations and unnecessary cost. Second, explanations generated by the existing explainers may not be deterministic or consistent for the same input graph over different runs, because each run of secondary model learning may lead to different outcomes. Moreover, a number of approaches require gradient back-propagation, and suffer from the saturation problem (Shrikumar et al., 2017), where a saturated gradient refers to a function for which a bigger input will not lead to a relevant increase in output. So if the gradient is saturated (which means it is extremely close to zero), a bigger upstream gradient does not lead to a bigger current gradient when applying the chain rule. We revisit the research question of GNN explanation and point out that there is no need to train auxiliary models or use gradient information for GNN explanation. Rather, by directly analyzing the GNN input-output relationship itself, GOAt is a training-free method without requiring any auxiliary model. GOAt forwardly attributes graph outputs to input graph features (and edges), naturally producing consistent explanations across different runs, and avoids relying on gradients. Moreover, GOAt demonstrates stronger fidelity, discriminability and stability through experiments on 6 datasets over 3 types of GNNs.
>
> **W2**: As we explained at the beginning of Section 3, we first expand the GNN output as a sum of scalar products involving input features (e.g. edges) and activation patterns. Then for each scalar product, we apply Equal Contribution concept. We have added a figure (Figure 1) illustrating our method in the updated manuscript (highlighted in red). As Figure 1 shows, we first expand the representation of GNN’s input-output relation into a sum of product terms. For example, the expansion form of an element in the output matrix can be $y_{13}=10A_{11}A_{12}A_{23}+8A_{11}A_{13}A_{33}+11A_{12}A_{21}A_{13}$. Then, for each product term, like $10A_{11}A_{12}A_{23}$, the elements in adjacency matrix $A$ are from {0,1}, indicating the absence or presence of edges between pairs of nodes. The absence of any edge among $A_{11}, A_{12}, A_{23}$ would make the whole term $10A_{11}A_{12}A_{23}$ equal to $0$. Therefore, we say that the three edges are equally important to this product term. Hence, all three edges evenly share the contribution to this term, where each edge contributes $10/3$. Finally, we investigate all the product terms, and obtain the contribution of each edge by summing up its contribution to all the product terms. For instance, the contribution of $A_{11}$ to output $y_{13}$ is $10/3+8/3=6$. The proof of Equal Contribution in a scalar product can be found in Appendix A.
>
> **W3**: As discussed in Sections 1 and 5, GLGExplainer and GCFExplainer are global-level explainers, whereas this paper focuses on instance-level GNN explainability. Therefore, they did not appear in the experiments as baselines.
>
> **W4**: We notice that the reviewer wrote in the summary that “Empirical results on three well-known datasets show the effectiveness of the proposed method”. In fact, we conducted experiments on 6 datasets, including 3 node classification datasets and 3 graph classification datasets. Major works in this domain test on 5~7 datasets, e.g., PGExplainer (NeurIPS'20) tested on 6 datasets, SubgraphX (NeurIPS'21) tested 5 datasets, RG-Explainer (NeurIPS'21) 6 datasets, RCExplainer (NeurIPS 2021) 7 datasets, DEGREE (ICLR 2022) 6 datasets, TAGE (NeurIPS 2022) 6 datasets. Furthermore, unlike most existing methods that were only tested on GCN, we additionally conducted experiments on more variants of GNNs (including GCN, GIN, GraphSAGE). We even re-ran all the baselines since they were not trained on GIN or GraphSAGE. We re-implemented some baselines, e.g., because author implementation of DEGREE suffers from memory leak, RCExplainer is not open-sourced, subgraphX is a costly search-based method, RCExplainer, RG-Explainer are RL methods requiring long training. These represent extensive and significant experimental efforts, comparable to or even more than the prior works. Moreover, during the rebuttal period, we ran additional experiments on BBBP, providing further evidence of our approach's efficacy (see Appendix L, highlighted in red). These additional results align with the discussion in Section 4, which shows that our method outperforms all baseline methods.
>
> **W5**: Thanks for pointing out the typo. We have removed it from the updated manuscript.
>
> We would appreciate it if the reviewer could re-assess the work. The gist here is that we show that GNN explanation really does not need to resort to gradient or another model. There is a surprisingly simple analytical way to attribute graph outputs to input features, which is also effective and beats SOTA methods.

---

### Official Review · Reviewer_8TnY · 2023-10-29

**Soundness:** 2 fair
**Presentation:** 2 fair
**Contribution:** 2 fair
**Rating:** 8
**Confidence:** 3

**Summary:**

This paper introduces GOAt, a novel method for GNNs by representing the output with product terms. Specifically, the authors propose an efficient analytical method to compute the contribution of each node or edge feature to each scalar product and aggregate the contributions from all scalar products in the expansion form to derive the importance of each node and edge. GOTt is shown to be applicable to different GNN architectures, including GCN, GraphSAGE, and GIN. It outperforms various SOTA GNN explanation baselines in terms of fidelity, discriminability, and stability as demonstrated through experiments on both synthetic and real-world data. The authors also provide qualitative case studies to show that GOAt can identify the most important nodes and edges in the input graph that contribute to the output prediction and that it can also detect the presence of adversarial attacks and the robustness of the GNN model.

**Strengths:**

1. Novelty: Explaining GNNs via output attribution is a novel approach to my knowledge.

2. Effectiveness: GOAt outperforms SOTA baselines in terms of fidelity, discriminability, and stability, and case studies show GOAt can generate qualitatively meaning explanations.

3. GOAt is shown to work for different GNN architectures and for both graph classification and node classification.

4. Code is provided for reproducibility.

**Weaknesses:**

1. Given the GOAt method is relatively complex, a time complexity analysis or empirical speed evaluation helps assess its significance.

2. For the newly introduced metrics, i.e., discriminability and stability, a more detailed discussion of how they are computed can help improve clarity.

**Questions:**

1. For the discriminability metric, which the class labels were used? The ground truth labels or the GNN predicted labels? In these two cases, the discriminability has different meanings. My current understanding is using the ground truth labels, but this means when GNN makes an incorrect prediction, the embedding is also included in the discriminability computation.

2. I am a bit confused about how the coverage for the stability measure is computed. Can the authors explain more? Also, although coverage for BA-2Motifs is high for GOAt, it is very low on Mutagenicity and NCI1 for all methods. Any discussions about this observation?

---

> ### Author Response · Authors · 2023-11-23
>
> **W1**: Thank you very much for the advice. We updated our manuscript by adding the empirical time analysis in Appendix K (highlighted in blue). We compared with the outstanding baselines, which are the perturbation-based method GNNExplainer (Ying et al., 2019), decomposition-based method DEGREE (Feng et al., 2022), and the search-based method SubgraphX (Yuan et al., 2021). Our approach is much faster than SubgraphX and GNNExplainer, and runs slightly faster than DEGREE while providing more faithful, discriminative and stable explanations as demonstrated in Section 4, which showcases the significance of our method.
>
> **W2**: We updated our manuscript by adding the formulas that compute discriminability and stability in Section 4.2 and 4.3.  (highlighted in blue).
>
> **Q1**: When we evaluate the discriminability, we filtered out the wrongly predicted data samples. Therefore, only when GNN makes a correct prediction, the embedding would be included in the discriminability computation. We updated our manuscript by adding this clarification in Section 4.2. (highlighted in blue)
>
> **Q2**: To clear up the confusion on the definition of coverage/stability, we have updated the manuscript with the formula to calculate stability in Section 4.3 (highlighted in blue).  The reason our approach shows high coverage for BA-2Motifs but relatively low coverage for Mutagenicity and NCI1 is rooted in the nature of these datasets. BA-2Motifs is synthetic, where the GNN achieves a 100% prediction accuracy (Table 2). This makes outstanding subgraphs, like the "house" motif, clearer to the original GNN, making it easier for our explainer to achieve high coverage with just a few explanation subgraphs. On the other hand, Mutagenicity and NCI1 are real-world datasets where the original GNNs achieve around 80% accuracy. This poses a challenge for the GNN itself to identify important subgraphs. Moreover, these real-world datasets involve more types of subgraphs, making it difficult to achieve high coverage with only a few explanation subgraphs. Therefore, the results from all baselines, including our method, are reasonable given these challenges.

---

### Official Review · Reviewer_PCGE · 2023-10-29

**Soundness:** 3 good
**Presentation:** 2 fair
**Contribution:** 3 good
**Rating:** 6
**Confidence:** 2

**Summary:**

The paper proposes a new GNN explanation method by expanding the GNN as a sum of scalar products. The proposed method efficiently computes the contribution of each node and feature. From extensive experiments, the paper shows significant improvements over baselines and also has better stability across samples.

**Strengths:**

1. The GNN explanation problem is important for decision-making.
2. The proposed method is sound and could explain both edges and features.
3. The proposed method significantly outperforms baselines.

**Weaknesses:**

1. There is no computation complexity analysis for the proposed method.
2. It is better to have a figure to illustrate the key idea of the proposed method.

**Questions:**

Please refer to the weaknesses.

---

> ### Author Response · Authors · 2023-11-23
>
> **W1**: Our primary goal is to introduce a training-free explanation algorithm that calculates the importance of graph components for each data instance in a forward analytical manner. While our main emphasis isn't on minimizing computation complexity, we've incorporated an empirical time cost analysis in Appendix K (highlighted in blue) to provide additional insights into the computational aspects of our approach. We compared with the outstanding baselines, which are the perturbation-based method GNNExplainer (Ying et al., 2019), decomposition-based method DEGREE (Feng et al., 2022), and the search-based method SubgraphX (Yuan et al., 2021). As an analytical method, our approach is much faster than SubgraphX and GNNExplainer, and runs slightly faster than DEGREE while providing more faithful, discriminative and stable explanations as demonstrated in Section 4, which showcases the significance of our method.
>
> **W2**: In response to the reviewer's suggestion, we have added a figure (Figure 1) illustrating our method in the updated manuscript (highlighted in red). As illustrated in Figure 1, we first expand the representation of GNN’s input-output mapping to a sum of product terms. For example, the expansion form of an element in the output matrix can be $y_{13}=10A_{11}A_{12}A_{23}+8A_{11}A_{13}A_{33}+11A_{12}A_{21}A_{13}$. Then, an example of a product term is $10A_{11}A_{12}A_{23}$. The elements in the adjacency matrix $A$ can take values from {0,1}, indicating the absence or presence of edges between pairs of nodes. The absence of any edge among $A_{11}, A_{12}, A_{23}$ would make the whole term $10A_{11}A_{12}A_{23}$ equal to $0$. Therefore, we say that the three edges are equally important to this product term. Hence, all the three edges evenly share the contribution of this term, where each edge has a contribution of $10/3$. Finally, we investigate all the product terms, and obtain the contribution of each particular edge by summing up its contribution to all the product terms. For instance, the contribution of $A_{11}$ to $y_{13}$ is $10/3+8/3=6$.

---

### Official Review · Reviewer_cqeJ · 2023-11-01

**Soundness:** 3 good
**Presentation:** 3 good
**Contribution:** 2 fair
**Rating:** 5
**Confidence:** 4

**Summary:**

This paper focuses on a problem of explaining graph neural networks. Specifically, the author proposed an explaining method that lies in the category of decomposed approaches. The proposed method aims to compute the contribution of each node/edge for the final predictions. Experiments are conducted to compare with some perturbation-based explanation methods.

**Strengths:**

1. This paper focuses on an important problem of explaining graph neural networks.
2. The proposed method is reasonable and technically sound.
3. The results compared with the representative perturbation-based explainers are promising.

**Weaknesses:**

1. There are already a group of decomposition methods for explaining graph neural networks [1, 2]. They share very similar idea with the proposed method. However, they are not discussed in the paper. It is necessary to clarify the contributions of this work compared with these related works.
2. In the experimental section, the decomposition methods are not compared with the proposed framework. It is suggested to add decomposition-based explainers as basslines.
3. It is claimed that the proposed method can be faithful and also stable across similar samples. I suggest to discuss and compare with a recent work [3] that also works in both directions of GNN explainer.

[1] Pope, Phillip E., et al. "Explainability methods for graph convolutional neural networks." Proceedings of the IEEE/CVF conference on computer vision and pattern recognition. 2019.
[2] Schnake, Thomas, et al. "Higher-order explanations of graph neural networks via relevant walks." IEEE transactions on pattern analysis and machine intelligence 44.11 (2021): 7581-7596.
[3] Zhao, Tianxiang, et al. "Towards Faithful and Consistent Explanations for Graph Neural Networks." Proceedings of the Sixteenth ACM International Conference on Web Search and Data Mining. 2023.

**Questions:**

Please refer to the weaknesses.

---

> ### Author Response · Authors · 2023-11-23
>
> **W1**: In Section 5, we already have discussed the methods mentioned by the reviewer, namely, Pope et al. (2019) and Schnake et al. (2021). Pope et al. (2019) propose several methods, namely, SA, CAM, Grad-CAM and Excitation-BP, where Excitation-BP is categorized as a decomposition method according to the survey paper [4]. On the other hand, Schnake et al. (2021) proposes a decomposition method GNN-LRP. These methods measure the importance of input features by decomposing the original model predictions into the importance scores of the corresponding input features. And what's important is that they all rely on gradient backpropagation to achieve this. As gradient-based methods, they face the saturation problem (Shrikumar et al., 2017), where a saturated gradient refers to a function for which a bigger input will not lead to a relevant increase in output. So if the gradient is saturated (which means it is extremely close to zero), a bigger upstream gradient does not lead to a bigger current gradient when applying the chain rule. Saturated gradients may potentially lead to less faithful explanations generated by these gradient-based methods.
>
> The novelty of our approach is not that it is a decomposition method (there are a bunch of decomposition methods already). Rather, the novelty of GOAt lies in that we directly attribute graph outputs to input graph features in an analytical way based on the notion of “equal contribution” without using gradients at all, thus not suffering from the issues these gradient methods are suffering from. That is, the idea proposed by this work is to offer a way to directly attribute GNN output to each input graph feature including edges. This is the major difference and novelty which depart from how the gradient-based methods derive explanations.
>
> [4] Yuan, Hao, et al. “Explainability in Graph Neural Networks: A Taxonomic Survey.” IEEE Transactions on Pattern Analysis and Machine Intelligence, 2022, pp. 1–19.
>
> **W2**: We did have compared our method with the leading decomposition method DEGREE (Feng et al., 2022) in our experiments already, which is a more recent decomposition method than Pope et al. (2019) and Schnake et al. (2021). Please check Section 4 and Appendix G for detailed insights. Through extensive experiments across 6 datasets and 3 variants of GNNs, our approach consistently demonstrates superior faithfulness, discriminability, and stability compared to the baseline methods.
>
> **W3**: Thank you for recommending a recent related work. We will cite and discuss this work. This paper also points out some issues of existing work that the explanations generated by some typical existing explanation methods may not be deterministic even for the same input graph, since they require training an auxiliary or secondary model; the auxiliary model may generate inconsistent explanations even for the same input graph over different runs of the model training. A lack of consistency will compromise the faithfulness of the explanation as well. To address this issue, they introduce alignment objectives to improve existing methods that require auxiliary model training, where the embeddings of the explanation subgraph and the raw graph are aligned via anchor-based objective function, distributional objective function based on Gaussian mixture models, and mutual information-based objective functions. In contrast, we propose a completely training-free instance-level explainer that does not need to be optimized to any learning objective. Being a novel analytical framework without resorting to an auxiliary model or training at all, GOAt naturally generates consistent explanations across different runs. Furthermore, GOAt offers the additional advantage of producing stable explanations even across similar graph samples as has been demonstrated in our experiments, proving the significance of our approach. We have added discussion on this related work in Section 5 in the updated manuscript (highlighted in green).
>
> We are committed to addressing reviewer concerns and improving the quality of this research. It is much appreciated if the reviewer could reassess the quality of this work. It is worth noting that we are not missing important related work and our analytical framework GOAt is fundamentally different from and performs better than gradient-based decomposition methods, as has been demonstrated in the paper.

---

### Meta-Review · Area_Chair_XZgJ · 2023-12-07

**Metareview:**

This paper introduces a method for explaining graph neural networks. The method works by attributing graph outputs to input graph features, overall aiming at creating transparent and deterministic explanations. The authors perform experiments on a variety of datasets and compare with other established GNN explaining methods.

__Strengths:__
- Useful, well-motivated approach. Also it is novel in terms of output attribution, aiming to provide more transparent explanations.
- Experiments are convincing
- Generality of the method

__Weaknesses:__
The reviewers have raised a few issues however I find that most have been addressed satisfactorily during the rebuttal period. The following two concerns of the reviewers have not been addressed 100%, but I do feel that the authors' clarifications have largely alleviated them:
- clarity
- need for deeper analysis on some aspects of the method

**Justification For Why Not Higher Score:**

The scores are not very high, the reviewers liked the paper but they were not particularly excited overall.

**Justification For Why Not Lower Score:**

This is a useful paper, and the rebuttal addressed most of the comments raised by the reviewers.

---

### Decision · Program_Chairs · 2024-01-16

Accept (poster)